# Proteomic and metabolomic analyses of the human adult myocardium reveal ventricle-specific regulation in end-stage cardiomyopathies
Benjamin Hunter [1,2,3,16], Mengbo Li [4,5,16], Benjamin L. Parker [6], Yen Chin Koay[1,2,3,7], Dylan J. Harney [2,8], Evangeline Pearson[9], Jacob Cao[10], Gavin T. Chen [11], Oneka Guneratne[12], Gordon K. Smyth [4,13], Mark Larance [2,3], John F. O'Sullivan [1,2,3,7,10,11,14,17] ✉ & Sean Lal [1,2,3,10,11,15,17] ✉

The left and right ventricles of the human heart are functionally and developmentally distinct such that genetic or acquired insults can cause dysfunction in one or both ventricles resulting in heart failure. To better understand ventricle-specific molecular changes influencing heart failure development, we first performed unbiased quantitative mass spectrometry on pre-mortem non-diseased human myocardium to compare the metabolome and proteome between the normal left and right ventricles. Constituents of gluconeogenesis, glycolysis, lipogenesis, lipolysis, fatty acid catabolism, the citrate cycle and oxidative phosphorylation were down-regulated in the left ventricle, while glycogenesis, pyruvate and ketone metabolism were up-regulated. Inter-ventricular significance of these metabolic pathways was then found to be diminished within end-stage dilated cardiomyopathy and ischaemic cardiomyopathy, while heart failure-associated pathways were increased in the left ventricle relative to the right within ischaemic cardiomyopathy, such as fluid sheer-stress, increased glutamine-glutamate ratio, and down-regulation of contractile proteins, indicating a left ventricular pathological bias.

The left ventricle's (LV) conical lumen and increased myocardial thickness relative to the right ventricle (RV) in the human heart is due to its developmental origin from the first heart field, whereas the RV is derived from the second heart field[1–5]. The separate origins are apparent when we consider the pathogenesis of congenital defects in which only one ventricle of the heart is affected, such as hypoplastic left or right heart syndrome and isolated non-compaction of the ventricular myocardium[6–13]. Even in a disease preluded by autosomal missense mutations, such as non-ischaemic dilated cardiomyopathy (DCM), pathological dilation and systolic dysfunction may occur in both ventricles or specifically the left ventricle, due to the different physiology of systemic and pulmonary output influencing the aetiology of the disease[14–17]. However, the approach to managing acute right ventricular dysfunction often differs to left ventricular dysfunction due to the RV's inability to adapt to increased afterload pressure whereas the LV is less capable adapting to volume overload[18–22]. Furthermore, right ventricular involvement can exacerbate LV-dominant cardiomyopathies and has been used as a measure of survivability for patients with heart failure (HF)[18,23,24]. Therefore, there is a need to understand the molecular differences between the left and right ventricles both in normal development and in pathological disease.

HF is accompanied by a reduction in fatty acid utilisation and an increase in glucose and ketone body utilisation as energy substrates, with further cardiometabolic remodelling identified in metabolic syndrome and ischaemic cardiomyopathy (ICM), a disease principally affecting the LV[25–34]. Despite this, there has been limited research as to what metabolic changes occur in one ventricle relative to the other in human HF phenotypes, let alone the comparative metabolic and energetic states in human left vs right ventricles absent of disease. Comprehensive metabolic ventricle-specific analyses with paired protein profiles have been performed in healthy mice[35]. However, to our knowledge, previous human inter-ventricular comparisons have been limited to small sample size chamber-specific transcriptional[36–40] and proteomic studies of which the latter featured either post-mortem healthy myocardium or diseased pre-mortem myocardium of varying severities[41,42].

Herein, we performed proteomic and metabolomic analyses on pre-mortem and age-matched adult myocardium to investigate and characterise ventricle-specific metabolism, and expression of contractile constituents, in non-pathological adults. Analyses were performed on successfully merged data from our previous study[43] and newly generated data to increase

statistical power while avoiding unwanted batch effects. These results were then compared to inter-ventricular analyses within LV-dominant cardiomyopathies, DCM and ICM, to identify perturbations in end-stage HF.

We found relative down-regulation of proteins which promote glycolysis, fatty acid oxidation (FAO), and lipogenesis including the pentose phosphate pathway (PPP) in the non-diseased LV vs the RV, while pyruvate and ketone metabolising proteins were up-regulated. In end-stage ischaemic cardiomyopathy, these differences were no longer represented but showed increased expression of proteins associated with pathological pathways in the LV vs the RV. The inter-ventricular molecular analyses within this study aides to fill a critical gap in our understanding of the metabolic differences between the human left and right ventricular myocardium and may be used to inform future therapeutic targets for heart failure processes in one or both the ventricles.

## Results

Sixty unique age-matched male and female human hearts (LV and/or RV) underwent proteomic and metabolomic analyses (25–27 donor, mean age 48 yo; 13–14 DCM, mean age 54–55 yo; 17–19 ICM, mean age 54 yo) (Fig. 1). Mass spectrometry data was acquired from two separate datasets (2018 and 2020, Supplementary Fig. 1). Donor and HF patient (ICM and DCM) demographic data are given in Table 1.

### Donor protein analysis

The proteomics data was normalised by the RUV-III algorithm to remove batch effects and improve data consistency (Supplementary Fig. 2)[44]. Multidimensional plots (MDS) revealed distinct clustering and separation of the non-pathological donor proteome from the HF proteome and visible separation of the donor LV from the RV (Fig. 2a). Of the 2793 proteins analysed, 193 proteins were up-regulated and 646 were down-regulated in the donor LV vs RV after false discovery rate (FDR) correction (Fig. 2b, c, Supplementary Data 3). Kyoto Encyclopedia of Genes and Genomes (KEGG) pathway analysis revealed 13 positively enriched pathways including pyruvate metabolism ($P = 2.54 \times 10^{-3}$) and propanoate metabolism ($P = 5.09 \times 10^{-3}$) in the donor LV vs RV (Fig. 2d). Thirteen negatively enriched pathways were identified including retinol metabolism ($P = 6.86 \times 10^{-4}$) and the PPP ($P = 0.0156$). A heatmap with dendrograms was used to visualise the top 100 differentially expressed (DE) proteins across all donor samples with chamber origin and sex indicated (Fig. 2e). Location-specific differential expression analysis was performed to reveal the sex differences in donors. Eukaryotic translation initiation factor 1 A Y-linked (EIF1AY, fold change, FC = 1.93, FDR = $2.61 \times 10^{-5}$), a protein encoded on the Y-chromosome, was found to be up-regulated in LV males. Conversely, there were 36 DE proteins in males vs females in the RV, of which 26 were up-regulated; notably, large (RPL30, RPL32, RPL36AL) and small (RPS29) ribosomal subunit proteins, and translational and trafficking proteins (HBS1L, EIF2A, MCFD2). Ten proteins were down-regulated in males vs females in the RV, notably SNTA1 (FC = 2.37, FDR = $4.52 \times 10^{-12}$), a cytosolic scaffolding protein which regulates cardiomyocyte SCN5A voltage-gated sodium channels and ATP2B4 calcium pumps[45,46], and CTNNA1 (FC = 2.31, FDR = $7.57 \times 10^{-6}$), a structural protein associated with intercalated discs and adherens junctions of cardiomyocytes[47] (Fig. 2f–h, Supplementary Data 6).

### Donor metabolite analysis

The metabolomics data was normalised by the RUV-III algorithm to remove batch effects and improve data consistency (Supplementary Fig. 3)[44]. MDS plots revealed an evident population separation of the donor metabolome from the HF metabolome and perceptible clustering of the RV relative to the LV (Fig. 3a). Of the 100 metabolites analysed, 10 were up-regulated and 15 were down-regulated in the LV vs RV (Fig. 3b, c, Supplementary Data 16). KEGG pathway analysis identified 5 pathways to be positively enriched including vitamin B6 metabolism ($P = 9.45 \times 10^{-3}$) and pyrimidine metabolism ($P = 0.0219$), whereas nicotinate and nicotinamide metabolism was negatively enriched ($P = 0.0487$) (Fig. 3d). A heatmap with

dendrograms was used to highlight differentially abundant (DA) metabolites with chamber origin and sex indicated (Fig. 3e). Sex-specific analysis revealed aspartic acid (FC = 2.17, FDR = $2.88 \times 10^{-3}$) to be up-regulated in the LV of males vs females, whereas fructose 6-phosphate (FC = 1.88, FDR = $2.88 \times 10^{-3}$) and erythrose 4-phosphate (FC = 1.78, FDR = 0.0296) were down-regulated. There were no DA metabolites within the RV between sexes (Fig. 3f–h).

### Heart failure protein analysis

In the LV vs RV of HF patients, 28 proteins were up-regulated and 121 were down-regulated (Fig. 4a, Supplementary Data 7). Seventy-nine proteins were found commonly DE between the donor and HF groups in the LV vs RV analyses, however, 4 were expressed in opposite directions such as ATPase Na$^+$/K$^+$ transporting Subunit alpha 1 (ATP1A1, FC = 0.323, FDR = $6.79 \times 10^{-3}$), which was up-regulated in HF LV (Fig. 4b, c, Supplementary Data 8). An MDS plot showed less separation of the LV from the RV of the DCM proteome compared to the donor group (Fig. 4d). Only 4 proteins were DE in the DCM LV vs RV, and were down-regulated, including acidic residue methyltransferase 1 (ARMT1, FC = 1.63, FDR = 0.035), which was also down-regulated in donor LV vs RV (Fig. 4e–g). The ICM proteome presented a detectable pattern of separation of the RV from the LV, however, less than the donors, as shown in the MDS plot (Fig. 4h). Proteomic analysis of ICM patients revealed 47 proteins were up-regulated and 184 were down-regulated in the LV vs RV (Fig. 4i, Supplementary Data 11). ICM LV vs RV KEGG pathway analysis revealed 15 pathways which were positively enriched including fluid sheer stress and atherosclerosis ($P = 0.0239$) and diabetic cardiomyopathy ($P = 0.0490$) (Fig. 4j). Two pathways were found to be negatively enriched: pyrimidine metabolism ($P = 0.0161$) and the PPP ($P = 0.0324$). Of the 231 proteins DE in the ICM LV vs RV, 112 proteins were commonly significant in the donor LV vs RV analysis, 7 of which (annotated in Fig. 4k) were DE in the opposite direction such as nitric oxide synthase 3 (NOS3, FC = 10.64, FDR = 0.0365) (Fig. 4l).

### Heart failure metabolite analysis

Metabolomic analysis in HF patients revealed 4 metabolites that were up-regulated including glutamic acid (FC = 1.09, FDR = $7.42 \times 10^{-3}$), and 1 down-regulated, deoxycholate (FC = 1.44, FDR = 0.0262), in the LV vs RV and (Fig. 5a). Adenosine was found to be up-regulated in the LV of HF patients (FC = 1.10, FDR = $7.42 \times 10^{-3}$) which was opposed to the down-regulation of adenosine in donor LV vs RV (FC = 1.11, FDR = $1.41 \times 10^{-3}$) (Fig. 5b, c). The DCM metabolome presented limited separation between the LV and RV (Fig. 5d). Acetoacetic acid was the only metabolite DA in the LV vs RV of DCM patients and was down-regulated (FC = 1.14, FDR = 0.0331), which was also down-regulated in the donor LV vs RV (Fig. 5e–g). The ICM metabolome appeared to cluster with the DCM metabolome, while also displaying no discernible dissociation between the LV and RV (Fig. 5h). Glutamic acid (FC = 1.11, FDR = 0.0212) and adenosine (FC = 1.11, FDR = 0.0212) were up-regulated in the LV vs RV of ICM, while deoxycholate was down-regulated (FC = 1.58, FDR = 0.0212), reflecting the changes seen within HF as a collective (Fig. 5i–k).

### Correlation network analyses

Correlation network plots were formed from the most significant DE proteins in the LV vs RV analyses for donors (200 proteins, Fig. 6a), DCM (4 proteins, Fig. 6b) and ICM (200 proteins, Fig. 6c). Significantly regulated metabolites were manually incorporated with their functionally relevant terms. Protein nodes were clustered into communities based on the absolute value of the correlation coefficient of LV and RV protein intensity where the adjoining proteins shared the highest correlation coefficient. Larger protein nodes were representative of a higher degree of correlation with and compared to other adjoining nodes. In the donor network plot, the largest community was cluster 5 which highlighted the co-regulation of many mitochondrial ribosomal proteins (MRP) relative to both the left and right ventricles. Proteins involved in glycolysis (ALDOC), glycogenesis (GYS1)

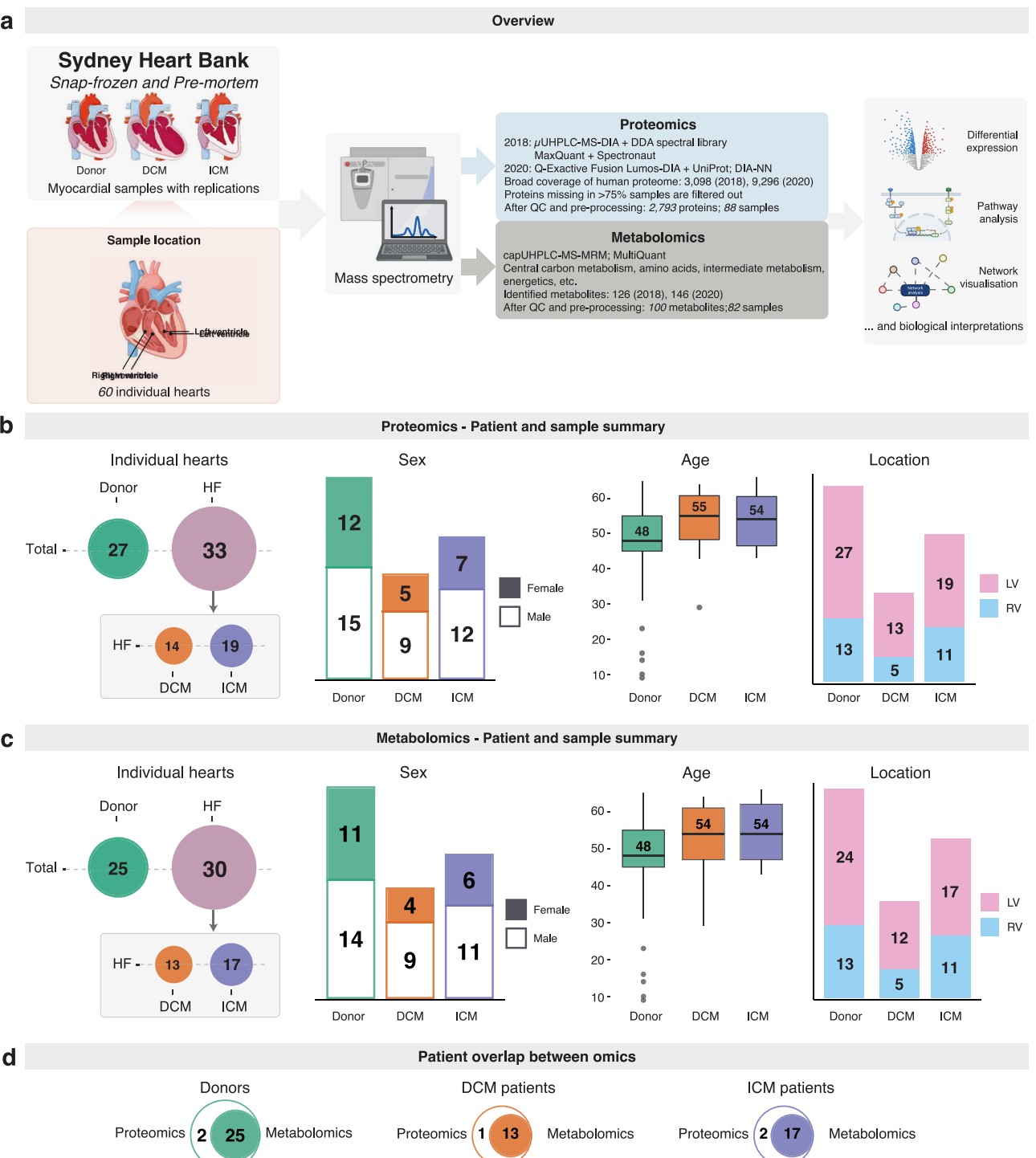

**Fig. 1 | Schematic summary. a** Schematic outlining sample acquisition, omic platforms applied and the downstream analysis workflow. **b** Patient and sample summary of the proteomics experiment. Top, from left to right: disease condition distribution in all measured hearts; sex counts in donor, DCM and ICM hearts; age distribution of donor, DCM and ICM hearts, where boxplots indicate the medians (the middle line), first and third quartiles (the box) and the whiskers show the 1.5× IQR above and below the box; and sample location distribution. Bottom: pie charts counting the number of donor, DCM or ICM hearts that have both LV and RV samples measured; only LV samples measured or only RV samples measured in proteomics. **c** Patient and sample summary of the metabolomics experiment. Top,

from left to right: disease condition distribution in all measured hearts; sex counts of donor, DCM and ICM hearts; age distribution of donor, DCM and ICM hearts by boxplots; and sample location distribution. Bottom: pie charts counting the number of donor, DCM or ICM hearts that have both LV and RV samples measured; only LV samples measured or only RV samples measured in metabolomics. **d** Counts of individual hearts measured by each omic platform by condition via Venn diagrams. Source data are provided in Supplementary Data 2 and 15. (DCM dilated cardiomyopathy; ICM ischaemic cardiomyopathy; IQR interquartile range; LV left ventricular; RV right ventricular).

**Table 1 | Demographics of non-pathological donors and heart failure patients**

| Characteristic | Donor | | DCM | | ICM | |
|---|---|---|---|---|---|---|
| | P.omics | M.omics | P.omics | M.omics | P.omics | M.omics |
| Count (n) | 27 | 25 | 14 | 13 | 19 | 17 |
| Age (years) (mean) | 45.0 | 44.6 | 53.1 | 52.5 | 53.1 | 53.4 |
| Male (%) | 55.6 | 56.0 | 64.3 | 61.5 | 63.2 | 64.7 |
| BMI (kg/m²) (mean) | 26.2 | | 23.6 | | 26.6 | 27.1 |
| Min LVEF (mean) | >55 | | 19.3 | 18.3 | 25.4 | 22.7 |
| LVEDD (mm) (mean) | <55 | | 75.4 | 77.3 | 65.8 | 66.2 |
| NYHA | N/A | | 3–4 | | 3–4 | |
| TPG (mmHG) (mean) | N/A | | 9.8 | | 11.4 | |
| Hypertension (%) | 3.7 | 4.0 | 14.3 | 15.4 | 26.3 | 29.4 |
| Hypercholesterolemia (%) | 11.1 | 12.0 | 14.3 | 15.4 | 26.3 | 23.5 |
| Kidney disease (%) | 0.0 | | 21.4 | 23.1 | 10.5 | 5.9 |
| Diabetes (%) | 0.0 | | 14.3 | 15.4 | 26.3 | 17.6 |
| LVAD (%) | N/A | | 14.3 | 15.4 | 5.3 | 5.9 |
| Alcohol (%) | 14.8 | 16.0 | 7.1 | 7.7 | 15.8 | 11.8 |
| Smoking (%) | 7.4 | 8.0 | 7.1 | 7.7 | 26.3 | 23.5 |
| Aldosterone antagonist (%) | 0.0 | | 42.9 | 46.2 | 26.3 | 23.5 |
| Beta blocker (%) | 0.0 | | 50.0 | 53.8 | 31.6 | 29.4 |
| ACE inhibitor (%) | 0.0 | | 35.7 | 38.5 | 31.6 | 29.4 |
| Family history of heart disease (%) | 3.7 | 4.0 | 35.7 | 30.8 | 10.5 | 5.9 |

Premortem left and/or right ventricular myocardial samples for proteomic and metabolomic mass spectrometry were acquired from 60 individuals. Clinical information was not available for every individual. *Donor* non-pathological heart donors, *DCM* end-stage dilated cardiomyopathy patients, *ICM* end-stage ischemic cardiomyopathy patients, *P.omics* samples used in proteomics analysis, *M.omics* samples used in metabolomics analysis, *BMI* body mass index, *LVEF* left ventricular ejection fraction, *LVEDD* left ventricular end-diastolic diameter, *NYHA* New York heart association functional classification, *TPG* trans-pulmonary pressure gradient, *LVAD* left ventricular assisting device implant, *ACE* angiotensin converting enzyme.

and glycogenolysis (AGL) were also found to be co-regulated in cluster 4. A version of the network plot with proteins highlighted as up and down-regulated has been supplied in Supplementary Fig. 4.

## Discussion

The inter-ventricular analyses in this study advance our understanding of the comparative molecular architecture of the left and right ventricles of the human heart. By comparing multi-omics analyses of non-pathological and HF LV vs RV, this study provides insight into normal (differential) LV and RV development as well as ventricle-specific changes in end-stage HF in which there is a pathological bias towards the LV in acquired (ICM) and (largely) genetic (DCM) forms of cardiomyopathy[32,33,48,49].

If we examine acetyl-CoA synthesis in the donor LV vs RV, the up-regulation of pyruvate dehydrogenases (PDHA1, PDHB) and lactate dehydrogenase B (LDHB), and the highly significant (FDR ≤ 0.001) up-regulation of dihydrolipoamide dehydrogenase (DLD) and dihydrolipoamide acetyltransferase (DLAT) indicates there may be up-regulation of acetyl-CoA synthesis via lactate and pyruvate metabolism, which has been suggested previously in healthy adults[50]. This was also accompanied by a down-regulation of proteins responsible for pyruvate dehydrogenase inhibition (PDPR), pyruvate conversion to acetate (ACYP1) and pyruvate-lactate transport out of the cell (SLC16A7, highly significant), and an up-regulation of pyruvate transport into the mitochondria via mitochondrial pyruvate carrier 1 (MPC1)[51]. Together, these differences infer an increased pyruvate availability and conversion to acetyl-CoA in LV relative to the RV in normal adult hearts, a process, when dysfunctional, may induce cardiomyocyte hypertrophy and systolic dysfunction[51–54]. A reduced synthesis of acetyl-CoA from acetoacetyl-CoA via oxidation of fatty acids (FAs) was indicated in the donor LV vs RV by the highly significant, forty-fold down-regulation of acyl-CoA dehydrogenase family member 11 (ACAD11), as well as the down-regulation of 3-hydroxyacyl-CoA dehydrogenases (EHHADH, highly significant; HADHA) and acetyl-CoA acyltransferases (HADHB, ACAA2), which

break down very long, long and medium-chain fatty acids. Ketone metabolism also appeared up-regulated as seen by the increased expression of key proteins promoting the metabolism of branched-chain amino acids to acetoacetic acid (DLD, DBT, MCCC2) and then to acetoacetyl-CoA (OXCT1) where paired with the down-regulation of acetoacetic acid, may indicate its utilisation for acetyl-CoA synthesis. The highly significant up-regulation of acyl-CoA synthetase short-chain family member 1 (ACSS1) may also indicate increased acetyl-CoA synthesis via acetone. The down-regulation of acetoacetic acid also in the DCM LV vs RV, and the up and down-regulation of ACSS1 and ACYP1, respectively, also in the ICM LV vs RV, do not strongly support a differential increase of ketone metabolism in the LV relative to the RV in end-stage heart failure as seen in donors but, given the well-understood shift of energy substrates seen in LV-dominant cardiomyopathies, suggests that metabolic remodelling may occur in both ventricles[26,29,55].

Regarding glycolysis in the LV vs RV of donors, all DE proteins between glucose 6-phosphate and pyruvate in the glycolytic pathway showed significant (GAPDH, PGAM1, PKM) and highly significant (PGAM2, ALDOA, ALDOC) down-regulation suggesting a relative reduction in glucose-dependent metabolism in normal LV. Reduced anabolic metabolism was also apparent by the highly significant down-regulation of proteins in the oxidative phase of the PPP (H6PD, PGD, and a fifty-three-fold reduction of DERA). Proteins in the non-oxidative phase of the PPP (TKT, TALDO1) were also down-regulated, including the rate-limiting glucose-6-phosphate dehydrogenase (G6PD), which would result in fewer carbons recycled back into glycolysis[56]. The suggested relative down-regulation of the PPP in the donor LV vs RV is also supported by the highly significant reduced abundance of erythrose 4-phosphate, an intermediate product of the PPP.

Up-regulation of glycogen elongation was indicated in the donor LV vs RV by the increased expression of hexokinase 1 (HK1) which is responsible for the intracellular trapping of glucose as glucose 6-phosphate (down-regulated, highly significant), and the highly significant increased expression

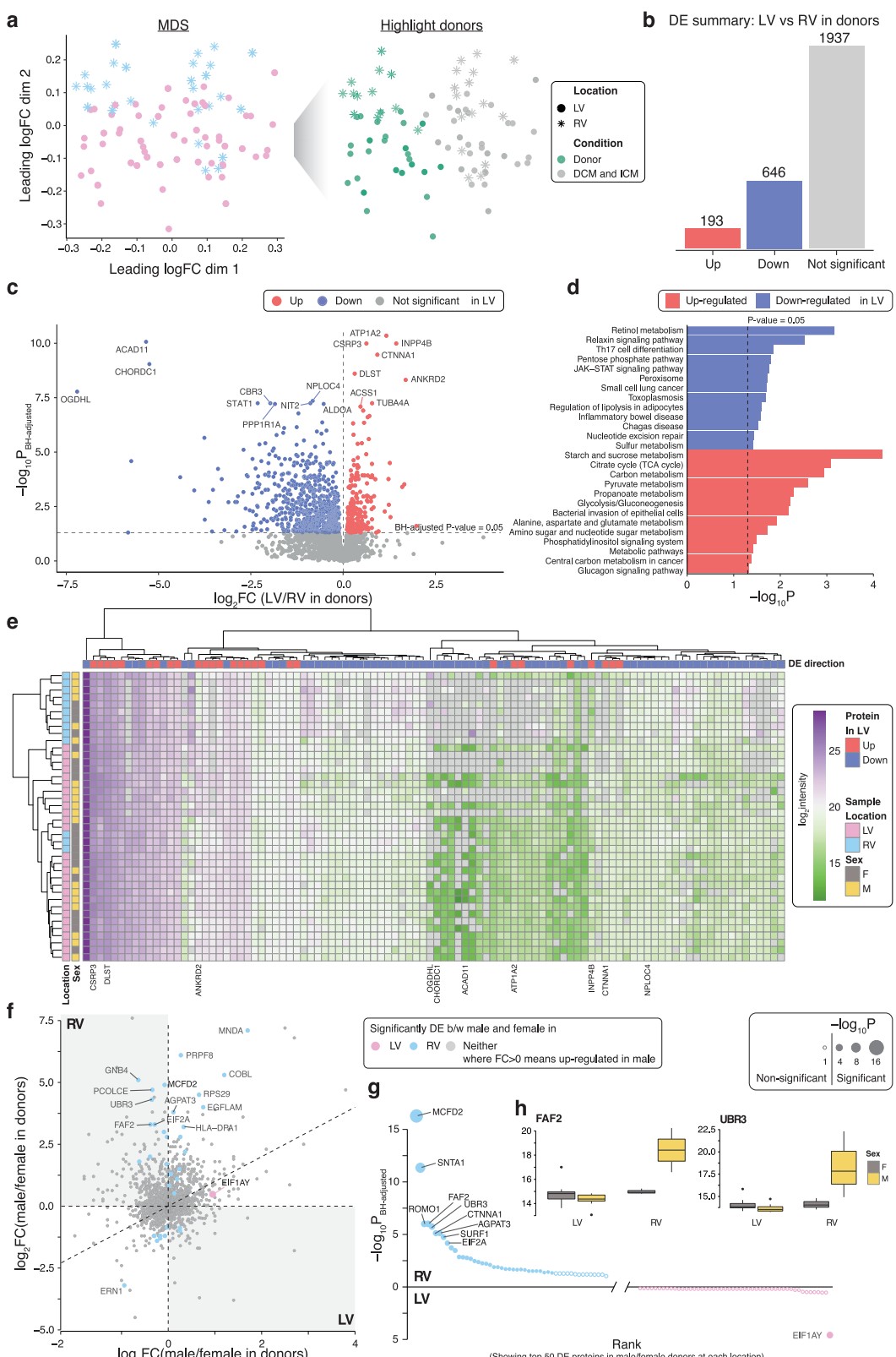

of critical proteins promoting the synthesis of glycogen via glucose 1-phosphate and UDP-glucose (PGM5, UGP2, GYS1). UDP, a substrate for the upstream synthesis of UDP-glucose, was also found up-regulated. This was further supported by the highly significant down-regulation of proteins which oppose glycogen synthesis (GALT, PYGM) and promote the reformation of glucose 1-phosphate during glycogenolysis[57–59].

With respect to FA synthesis in the donor LV vs RV, retinol (and beta-catenoid) metabolism, and associated proteins, which have been implicated in promoting lipogenesis showed significant (ADH1B, RBP7, RETSAT, ALDH1A1) and highly significant (RDH11, RBP4, ALDH1A2) down-regulation[60,61]. Solute carrier family 25 member 1 (SLC25A1), which is responsible for the export of citrate out of the mitochondria, and ATP citrate

**Fig. 2 | Differential analysis in protein expression levels between left and right ventricles and sex differences at each ventricle in donor hearts. a** MDS plot of proteomic data, a dimension reduction tool to visualise high dimensional data, where samples from donor hearts are highlighted in green on the right panel. **b** Summary of DE test results in proteins to compare LV with RV samples from donors. Estimates are derived using a linear regression model adjusted for sex and log2-transformed age, with $n_{Donor,LV} = 27$ and $n_{Donor,RV} = 13$. Direction of regulation is in reference to RV, that is, up-regulated proteins are higher expressed in LV than in RV samples. **c** Volcano plot of LV versus RV proteins in donors. **d** KEGG pathway analysis of proteins up-regulated or down-regulated in LV compared to RV samples in donors. Enriched KEGG pathways at $P$-value ≤ 0.05 are shown. **e** Heatmap with dendrograms of top 100 DE proteins in donor LV and RV samples.

Samples are annotated by sex and location. Proteins are annotated by direction of DE. **f** Plot of log2 FCs of proteins in female versus male donors in LV (x-axis) and RV (y-axis) samples. Direction of regulation is in reference to males, meaning that up-regulated proteins (log2 FC > 0) are higher expressed in males. **g** Top 50 proteins in female versus male donors at each ventricle, ranked by negative log10 $P$-value from differerntial analysis comparing between sexes at each location, illustrating the protein-sex interaction in donors. **h** Boxplots showing the distribution of log2 intensities for FAF2 (left) and UBR3 (right) in donor hearts stratified by location and sex. Source data are provided in Supplementary Data 1, 3–6. (BH Benjamini-Hochberg; DE differential expression; FC fold change; LV left ventricular; MDS multidimensional scaling; RV right ventricular).

lyase (ACLY), an initiator and regulator of lipid and cholesterol synthesis[62,63], were also found to be down-regulated. Upstream of acetyl-CoA, it was inferred that cholesterol synthesis via the mevalonate pathway may be down-regulated as seen by the apparent down-regulation of NADPH synthesis via the PPP, the reduced differential abundance of mevalonate, and the differential expression of phosphomevalonate kinase (PMVK)[64,65]. FA synthesis and triacylglycerol synthesis via glycerol metabolism were also suggested to be down-regulated due to the reduced expression of monoglyceride lipase (MGLL) and 1-acylglycerol-3-phosphate O-acyltransferase 3 (AGPAT3, highly significant), and by the increased expression of acylglycerol kinase (AGK), which opposes FA and triacylglycerol synthesis. Additionally, AGPAT3 showed highly significant up-regulation in the RV of donor males vs females, indicating a greater inter-ventricular difference in non-pathological males.

Regarding FA catabolism in the LV vs RV of donors, proteins which initiate and regulate lipolysis (FABP4, FAF2, respectively, highly significant), and transporters which facilitate lipid uptake into the cell (LRP1, highly significant; VLDLR, SLC27A6) were found to be down-regulated. These proteins also regulate the peroxisome proliferator-activated receptor (PPAR) pathway which in turn regulates FA transport via proteins such as diazepam binding inhibitor (DBI), and FAO via proteins such as sterol carrier protein 2 (SCP2)[66], which were also found to be down-regulated in the donor LV. Fas-associated factor family member 2 (FAF2), an endoplasmic reticulum membrane protein which inhibits adipose triglyceride lipase-mediated lipolysis within the PPAR pathway, presented greater differential expression in the LV vs RV within donor males due to an apparent up-regulation in the RV of males vs females[67,68].

The peroxisome pathway, which was negatively enriched in the donor LV vs RV KEGG analysis, has also been shown to regulate insulin resistance and glucose uptake[69]. In addition to this, the highly significant down-regulation of free carnitine in donor LV vs RV and the down-regulated carnitine palmitoyltransferase 1A (CPT1A) in female donor LV vs RV may indicate relatively reduced transport of long-chain acyl-CoA into the mitochondria for FAO. Furthermore, the up-regulation of malonyl-CoA in the donor LV vs RV may also play a contributing role in inhibiting CPT1 forming acyl-carnitine for long-chain FA transport into the mitochondria[70]. CPT1A was not DE between the LV and RV in donor males after FDR correction but showed a down-regulation in the RV when compared to females. Interestingly, CPT1A was previously found to be down-regulated in the LV vs RV of healthy male mice[35].

Inner mitochondrial transport of short and medium-chain acylcarnitines was likely down-regulated in the donor LV vs RV by the reduced differential expression of SLC25A20 and acyl-CoA synthetase family member 2 (ACSF2), and the differential abundance of butyrylcarnitine. Carnitine acetyltransferase (CRAT), which regulates acetyl-CoA availability in the mitochondria by transferring the acetyl group to carnitine, was additionally down-regulated. This may be associated with a reduced abundance of acetylcarnitine favouring mitochondrial acetyl-CoA availability.

Of these FA metabolism protein characteristics in the donor LV vs RV, only LDL receptor-related protein 1 (LRP1) and fatty acid binding protein 4 (FABP4) were also reduced in ICM LV vs RV. It follows that FAO decreases

within the LV of human failing hearts[26,34,71], but herein it is decreased in the LV of non-pathological human hearts compared to the RV.

Considering the citrate cycle & NADH synthesis in the donor LV vs RV, KEGG pathway analysis identified the citrate cycle was positively enriched due to the relative increase in numerous constituent proteins, however, rate-limiting enzymes which produce NADH, an electron donor for oxidative phosphorylation, showed significant (IDH2) and highly significant (IDH1, OGDHL) reductions, inferring down-regulation of the pathway[72–74]. The up-regulation of glutamic-oxaloacetic transaminase 2 (GOT2) and glutamate dehydrogenase 1 (GLUD1) in the mitochondria, and the highly significant down-regulation of glutamic--pyruvic transaminase (GPT) in the cytosol suggests an increased activity to shunt α-ketoglutarate, synthesised from oxaloacetate (down-regulated) and glutamine (highly significant up-regulation) via glutamate, back into the citrate cycle[75]. Furthermore, the increased glutamine may suggest a relative increase in amino acid metabolism in the LV providing more carbons to the citrate cycle while there is a relative decrease in carbohydrate and fatty acid metabolism. Dihydrolipoamide S-succinyltransferase (DLST, highly significant), which finalises the synthesis of succinyl-CoA from α-ketoglutarate after the action of oxoglutarate dehydrogenase, was up-regulated. The suggested reduced synthesis of NADH agrees with the suggested reduced synthesis of NAD$^+$ from nicotinamide D-ribonucleotide as seen by the down-regulation of nicotinamide nucleotide adenylyltransferase 1 (NMNAT1), niacinamide (highly significant), and nicotinamide metabolism, while NAD$^+$ consumers were also down-regulated (PARP1, SIRT2). Nicotinamide N-methyltransferase (NNMT), which converts niacinamide to N$^1$-methylnicotinamide was also found to be down-regulated in the donor LV vs RV.

Of the above DE proteins in donor LV vs RV, only DLST was also up-regulated of the citrate cycle in the ICM LV vs RV, however, less so compared to the donor LV. In the ICM LV vs RV, glutamic acid was found up-regulated, however, it was found that the relative glutamine to glutamic acid ratio (Gln/Glu) was down-regulated suggesting an increased supply and utilisation of glutamate in the mitochondria. A reduced Gln/Glu has been shown to be a feature of heart failure and associated with obesity, insulin resistance, type-2 diabetes, and coronary artery disease in plasma studies[75–79]. This likely suggests that the more pronounced structural and functional changes of the LV, compared to the RV, in end-stage heart failure are mirrored at the metabolic level[43].

Regarding oxidative phosphorylation, differentially expressed proteins of complex I of the electron transport chain (NDUFA4, NDUFA10, NDUFB3, NDUFB6, NDUFB10, NDUFC1, NDUFS1, NDUFS2 and NDUFS3), as well as complex V (ATP12A, ATP5PF, ATP6AP1, ATP6), were all found to be down-regulated in the donor LV vs RV. Conversely, ubiquinol-cytochrome C reductase, complex III subunit XI (UQCR11) of complex III showed highly significant up-regulation in donor LV. No complex I proteins were DE within LV vs RV of HF, but complexes III (UQCRB, UQCRFS1) and V (ATP6V1F) were also up and down-regulated in the ICM LV vs RV, respectively. A decreased oxidative phosphorylation in end-stage heart failure due to cardiometabolic dysfunction has been previously described[80], however, here it has been shown that there is less inter-ventricular differential expression of the electron transport chain in

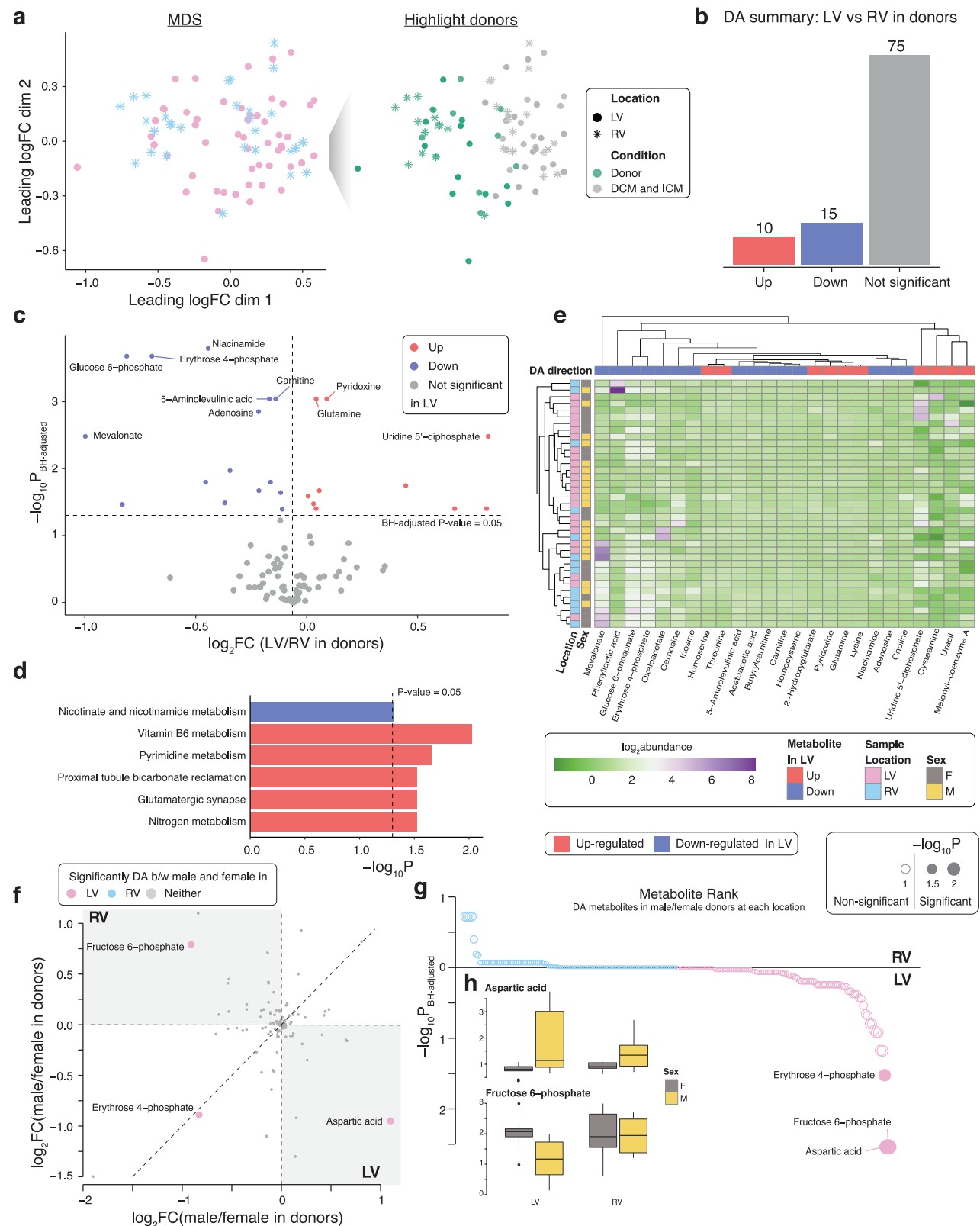

HF than donor hearts. A summary cellular schematic of suggested non-pathological up and down-regulated pathways in donor LV vs RV, informed by our multi-omics analysis is depicted in Supplementary Fig. 5.

There were also important changes in contractile and structural proteins, such as subunits of the sodium/ potassium ATPase, which were DE in the donor LV vs RV where ATP1A1 and ATP1B1 were found to be down-regulated, however, ATP1A2 (highly significant) was found to have a greater up-regulated fold change. Furthermore, ATP1A2 showed up-regulation in the RV of donor males vs females, indicating a reduced inter-ventricular difference in non-pathological males. In the ICM LV vs RV, the opposing up-regulation of ATP1A1 resulted in an observed increase in the cardiac muscle contraction KEGG pathway. Interestingly, ATP1A1, among other

**Fig. 3 | Differential analysis in metabolite abundance levels between left and right ventricles and sex differences at each ventricle in donor hearts. a** MDS plot of metabolomic data, a dimension reduction tool to visualise high dimensional data, with samples from donor hearts highlighted in green on the right panel. **b** Summary of DA test results in metabolites to compare LV with RV samples in donors. Estimates are derived using a linear regression model adjusted for sex and log2-transformed age, with $n_{Donor,LV}$ = 24 and $n_{Donor,RV}$ = 13. Direction of regulation is in reference to RV, that is, up-regulated metabolites are more abundant in LV than in RV samples. **c** Volcano plot of LV versus RV metabolites in donors. **d** KEGG pathway analysis of metabolites up-regulated or down-regulated in LV compared to RV samples in donors. Enriched KEGG pathways at $P$-value ≤ 0.05 are shown. **e** Heatmap with dendrograms of DA metabolites in donor LV and RV samples.

Samples are annotated by sex and location. Metabolites are annotated by direction of DA. **f** Plot of log2 FCs of metabolites in female versus male donors in LV (x-axis) and RV (y-axis) samples. Direction of regulation is in reference to males, meaning that up-regulated metabolites (log2 FC > 0) are more abundant in males. **g** Metabolites in female versus male donors at each ventricle, ranked by negative log10 $P$-value from differential analysis comparing between sexes at each location, illustrating the metabolite-sex interaction in donors. **h** Boxplots showing the distribution of abundances for aspartic acid (left) and fructose 6-phosphate (right) in donor hearts stratified by location and sex. Metabolite abundances are normalised to pooled samples. Source data are provided in Supplementary Data 14, 16–19. (BH Benjamini-Hochberg; DA differentially abundant; FC fold change; LV left ventricular; MDS multidimensional scaling; RV right ventricular).

ion channels, has been shown to be decreased in end-stage HF vs non-pathological hearts[81]. Hence, these inter-ventricular differences within donor and HF may not be equivalent to a direct comparison between the conditions. ATPase sarcoplasmic/endoplasmic reticulum $Ca^{2+}$ transporting 2 (ATP2A2), was down-regulated in the donor LV vs RV but was not DE between the ventricles of HF. Sarcomeric proteins myosin heavy chain 6 (MYH6), alpha-actinin 2 (ACTN2) and (CTNNA1), an adherens protein of the intercalated disc, were found to be up-regulated in the donor LV vs RV, whereas MYH8 was down-regulated in both the donor and ICM LV vs RV. It was previously found that the RV is more enriched in the vCM2 cardiomyocyte subpopulation and features a higher expression of MYH6 which validates our finding[40].

Interestingly, CTNNA1 showed highly significant down-regulation in the RV of donor males vs females, indicating a greater inter-ventricular difference in non-pathological males which may agree with a previous finding that cardiomyocytes are more abundant in females[40]. The non-pathological nineteen-fold up-regulation of ubiquitin-protein ligase E3 component N-recognin 3 (UBR3) in the RV of donor males vs females was also of interest as it is involved with the regulation of cardiomyocyte voltage-gated $Na^+$ channels (depolarisation) via the ubiquitin-proteasome pathway[82].

Matrix metallopeptidase 9 (MMP9), a collagenase which digests type IV and V collagen of the extracellular matrix which has been identified as a biomarker for coronary artery disease, was found to be down-regulated in both the LV of donors and ICM vs RV[83]. In addition to this, myomesin 3 (MYOM3) and titin (TTN), a protein which spans the length of, and regulates the contraction of the sarcomere, were down-regulated in the ICM LV vs RV. These proteins have previously been shown to be reduced and implicated in HF, particularly in DCM and ICM compared to non-pathological hearts[15,84–88], and so their reduction in the ICM LV vs RV, but not within DCM, is consistent with chronic ischaemia primarily inducing maladaptive remodelling in the LV rather than the RV, where nonsense genetic mutations often causing DCM affects both ventricles[89,90].

Regarding signalling, proteins critical in mTOR signalling (RPS6KA3, AKT2, MTOR, EIF4E) were found to be reduced in donor LV vs RV and were not DE within HF. The ICM LV vs RV KEGG protein analysis revealed up-regulation of fluid sheer stress and diabetic cardiomyopathy principally due to the up-regulation of endothelial nitric oxide synthase 3 (NOS3, seven-fold), caveolin 1 (CAV1). This paired with the down-regulation of the PPP (producer of NADPH which protects against oxidative stress) and the up-regulation of VEGF-signalling in the KEGG protein analysis likely indicates increased oxidative stress in the LV relative to the RV, and nitric oxide-mediated vasodilation in the coronary arteries, which is consistent with ischaemia primarily affecting LV rather than the RV[91–96]. The LV vs RV in donors showed a down-regulated trend in these pathways due to opposing differential expression of NOS3, CAV1, and CAV2 and reduced MAPK signalling proteins (MAP2K4, MAPK14).

This study stands apart from research before it in two distinctive ways: 1, by dissecting the left ventricular intramyocardial proteome and metabolome in contrast to the RV within healthy and diseased hearts; 2, of human origin where the tissue is not post-mortem. There are limitations to

our work. In contrast to a previous study analysing the proteome in regions of the heart and great vessels[97], we did not aim to capture an equivalent number of low-abundant proteins but to highlight the novelty of a ventricle-focused multi-omics analysis within and between non-diseased and diseased human myocardium. Despite the differential expression of many proteins and metabolites supporting common trends in metabolism, primary human myocardial cultures and animal models would be needed to confirm the cause and effect of the pathways that we have identified. Lastly, as the methods of this study could not distinguish between cell types, it was assumed significant differences were represented by cardiomyocytes (~56% of cells, ~85% volume)[40,98].

To our knowledge, this is the most comprehensive analysis of protein expression and metabolite abundance in human myocardium that infers differences between the LV and the RV in donor (non-diseased) hearts, and pathological differences between these chambers within two common heart failure phenotypes. In the non-pathological heart, our results infer that normal LV metabolism involves relatively lower glycolysis, glycogenolysis, pentose phosphate pathway metabolism, fatty acid transport and metabolism, lipogenesis, the citrate cycle, and oxidative phosphorylation. Conversely, compared to the RV, the LV had higher rates of glycogenesis, pyruvate, amino acid, and ketone metabolism. In comparison, these metabolic and energetic differences between the LV and RV were diminished in HF, in particular, DCM, which has important implications when we consider future therapeutic targets for homogenous versus heterogeneous heart failure processes that either equally or uniquely affect either or both ventricles.

## Methods
### Acquisition of human myocardium
Human LV myocardium was procured from the anterior wall and RV myocardium from the lateral wall, of (non-ischaemic) DCM and ICM patients expressing end-stage heart failure (NYHA III-IV), and non-pathological donors by the Sydney Heart Bank at St Vincent's Hospital Sydney as previously described[43,99–107]. Heart failure samples were extracted from the hearts of recipients undergoing heart transplantation. Non-pathological samples were acquired from donor hearts which could not be viably used for heart transplantation due to compatibility and logistical limitations at the time. Informed consent was obtained prior to the collection of all tissue. All donors of human myocardium were Caucasian. This was not due to selection but a result of tissue availability. Both the donor and HF hearts were delivered perfused and cardiopleged on wet ice. The myocardium was then concentrically cut from the outer wall, immediately snap-frozen in liquid nitrogen (−196 °C) on-site and stored long-term at −192 °C. Epicardium was excluded. Visibly identified myofibrotic scar tissue was avoided at infarct-affected areas during the collection from ICM hearts. All samples were collected within 40 min after aortic cross-clamp during surgery such that the tissue was not post-mortem, in keeping with the high-quality extracted RNA[99,105,108]. The absence/presence of pathology (donor vs HF, respectively) was verified histologically by hospital anatomical pathology. Specifically, donor hearts did not have disease, whilst the HF samples had confirmed disease in both ventricles. The methods of harvesting, storage and use of donated human myocardium were approved by the Human Research Ethics Committee at The University of Sydney

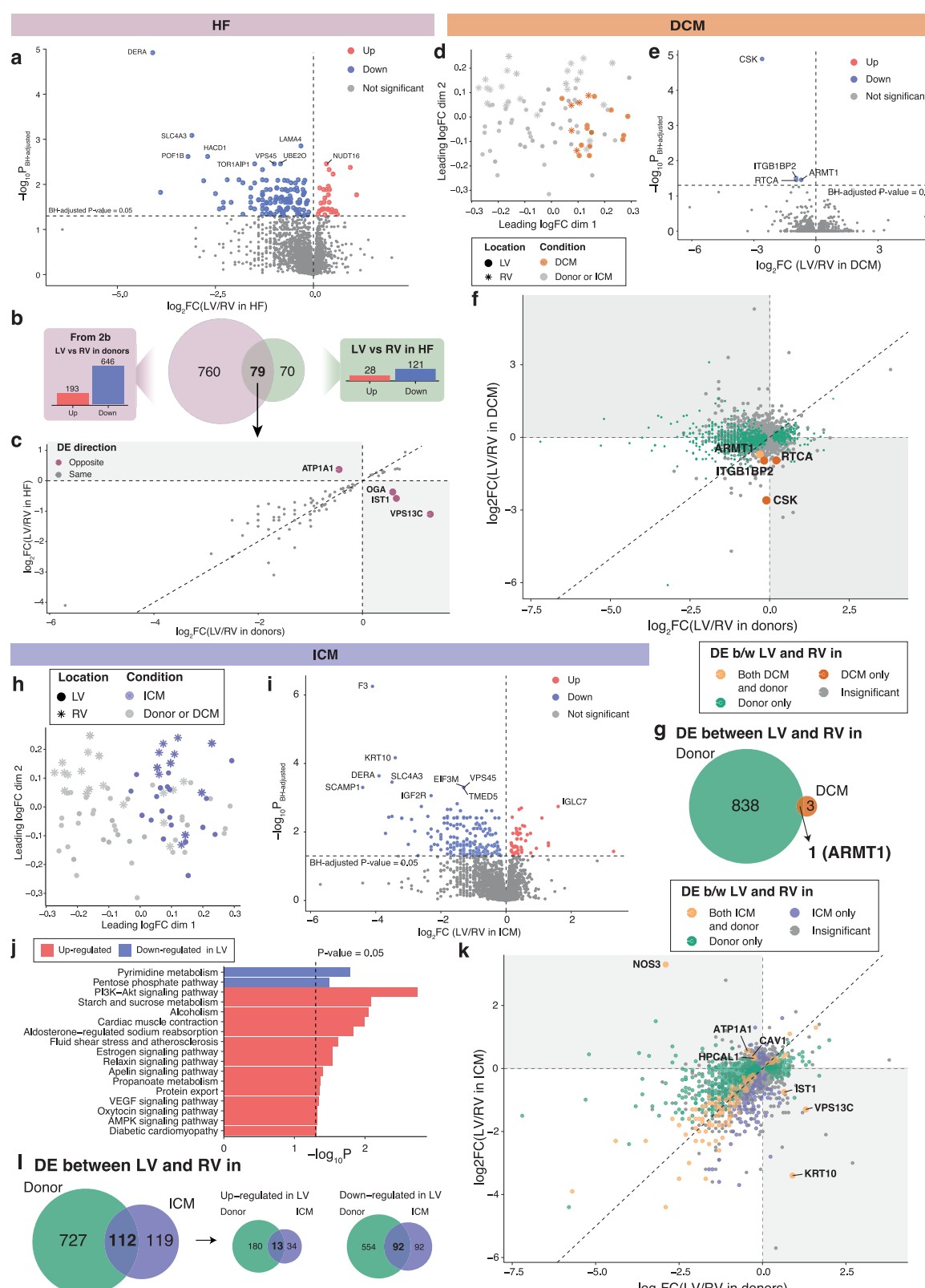

(USYD 2021/122). All ethical regulations relevant to human research participants were followed.

This study utilised data from samples which were used across two separate proteomic and metabolomic mass spectrometry analyses performed in 2018 and 2020 (Supplementary Fig. 1). This study featured 12 patients from the 2018 analyses (4 healthy donor, 4 DCM, and 4 ICM)

and 21–22 patients from the 2020 analyses (11–12 healthy donor and 10 ICM) which contributed both LV and RV samples, resulting in 28–27 individuals (13–12 donor, 4 DCM, 11 ICM) individuals contributing both an LV and an RV sample across both cohorts of proteomic and metabolomic mass spectrometry analyses, respectively (Supplementary Data 26–29).

**Fig. 4 | Differential analysis in protein expression levels between ventricles in heart failures (HF), DCM or ICM compared with donors. a** Volcano plot of LV versus RV proteins in heart failures. Estimates are derived using a linear regression model adjusted for sex and age, log2-transformed, with $n_{HF,LV}$ = 32 and $n_{HF,RV}$ = 16. Direction of regulation is in reference to RV, where up-regulated proteins are higher expressed in LV than in RV samples. **b** Venn diagram illustrating overlaps in DE proteins between ventricles in donors and in HF. **c** Plot of log2 FCs of DE proteins between ventricles in both donors and HF. **d–g** Differential analysis in protein expression levels between left and right ventricles in DCM compared with donors, where $n_{DCM,LV}$ = 13 and $n_{DCM,RV}$ = 5, adjusted for sex and age, log2-transformed. Direction of regulation is in reference to RV. **d** MDS plot of proteomics data with DCM samples highlighted in brown. **e** Volcano plot of LV versus RV proteins in DCM. **f** Plot of log2 FCs of proteins in LV versus RV samples in donors against those

of DCM. **g** Venn diagram counting the number of DE proteins between ventricles shared by/unique to donors and/or DCM. **h–l** Differential analysis in protein expression levels between left and right ventricles in ICM compared with donors, where $n_{ICM,LV}$ = 19 and $n_{ICM,RV}$ = 11, adjusted for sex and age, log2-transformed. Direction of regulation is in reference to RV. **h** MDS plot of proteomics data with ICM samples highlighted in purple. **i** Volcano plot of LV versus RV proteins in ICM. **j** KEGG pathway analysis of proteins up-regulated or down-regulated in LV compared to RV samples in ICM. Enriched KEGG pathways at $P$-value ≤ 0.05 are shown. **k** Plot of log2 FCs of proteins in LV versus RV samples in donors against those of ICM. **l** Venn diagrams counting numbers of all DE proteins (left); up-regulated DE proteins (middle); and down-regulated DE proteins (right) between ventricles shared by/unique to donors and/or ICM. Source data are provided in Supplementary Data 1, 7–13.

## Proteomics sample preparation and quantification

The 2018 proteomic mass spectrometry quantification was performed as previously described[109]. The 2020 proteomic mass spectrometry quantification was performed as follows: Cryopreserved human myocardium was powderised in liquid nitrogen and weighed to ~10 mg. The powder was homogenized in 4% sodium deoxycholate and 100 mM Tris-HCl pH 7.5. Samples were heated to 95 °C before sonication using QSonica R2 (Q Sonica) at 70% amplitude. Samples were clarified at $18,000 \times g$ and the pellet was discarded. Protein concentration was determined by BCA assay (Pierce). Samples were digested with trypsin overnight at 37 °C. Peptides were prepared for mass spectrometry as described previously[110]. A pooled sample was generated from these peptides which was separated offline with high-pH RP fractionation. Peptides were fractionated using an acquity UPLC M-Class CSH C18 130 Å pore size, 1.7 µm internal diameter and 300 µm × 150 mm column (Waters). The separation used buffer A which contained 2% acetonitrile and 10 mM ammonium formate (pH 9) and buffer B which contained 80% acetonitrile and 10 mM ammonium formate (pH 9). Separation occurred over 30 min with 96 fractions concatenated into 16 wells which were then dried and resuspended in 5% formic acid for injection.

Peptide samples were directly injected onto a 30 cm × 70 um C18 (Dr Maisch, Ammerbuch, Germany, 1.9 µm) fused silica analytical column with a 10 µm pulled tip, coupled online to a nanospray ESI source. The separation used buffer A; 0.1% formic acid in mass-spectrometry grade water and buffer B; 80% acetonitrile and 0.1% formic acid in mass-spectrometry grade water. Peptides were resolved over a gradient from 5% to 40% buffer B over 120 min with a flow rate of 300 nL min$^{-1}$. Peptides were ionized by electrospray ionization at 2.3 kV. MS/MS analysis was performed using a Q-Exactive Fusion Lumos mass spectrometer (ThermoFisher) with 27% normalised HCD collision energy for fragmentation. Spectra were attained in a data-independent acquisition using 20 variable isolation windows. RAW data files including the high pH fractions were analysed using the integrated quantitative proteomics software DIA-NN[111] (version 1.7). The database provided to the search engine for identification contained the Uniprot human database downloaded on the 5th of May, 2020. FDR was set to 1% of precursor ions. Both remove likely interferences and matches between runs were enabled. Trypsin was set as the digestion enzyme with a maximum of 2 missed cleavages. Carbamidomethylation of Cys was set as a fixed modification and oxidation of Met was set as variable modification. Retention time-dependent profiling was used and the quantification setting was set to any LC (high accuracy). Protein inference was based on genes. The neural network classifier was set to double-pass mode. The MaxLFQ algorithm was used for label-free quantitation, integrated into the DIA-NN environment[112,113].

## Metabolomics sample preparation and quantification

The 2018 metabolomic mass spectrometry quantification was performed as previously described[109], as was the 2020 quantification with changes mentioned in the following methods: Cryopreserved human myocardium was powderised in liquid nitrogen and weighed to ~50 mg. The tissue was lysed and homogenised in methanol/chloroform (2:1; v/v, HPLC grade). Equal

volumes of water and chloroform were added to the samples which were then centrifuged at 14,000 rpm at 4 °C for 20 min. The aqueous-containing metabolite layer was extracted and concentrated in the Speed-Vac SPD120 (Thermo Fisher Scientific) then dried under nitrogen stream, followed by reconstitution in the acetonitrile/methanol/formic acid (75:25:0.2; v/v/v, HPLC grade; Thermo Fisher Scientific) for the HILIC analysis, and acetonitrile/methanol (25:25; v/v/v, HPLC grade) for the AMIDE analysis. Deuterated internal standards were used for targeted metabolite profiling to determine mass spectrometry (MS) multiple reaction-monitoring transitions, declustering potentials, collision energies and chromatographic retention time, as described previously[114,115]. Liquid chromatography-tandem mass spectrometry (LC-MS/MS) system was used for both HILIC and AMIDE analyses and composed of an Agilent 1260 Infinity liquid chromatography (Santa Clara, CA, USA) system coupled to a QTRAP5500 mass spectrometer (AB Sciex, Foster City, CA, USA). Polar metabolites in both positive and negative ionisation modes were separated in hydrophilic interaction liquid chromatography (HILIC) mode using an Atlantis® HILIC column (Waters) and an XBridge™ Amide column (Waters), respectively, which allow the separation of metabolites of different properties, both as previously described[114,116]. The sample order was randomised across three batches. The analysis software SCIEX OS (AB SCIEX, version 1.7.0.36606) was used for multiple reactions monitoring Q1/Q3 peak integration of the raw data files. The abundance was quantified as the peak area of a metabolite which was then normalised to their closest bookended pooled myocardium extracts which were included after every 10 study samples in the sample queue, in order to adjust for any temporal drift in instrument performance.

## Statistics and reproducibility

All statistical analyses were performed in R[117](4.1.1). For both metabolomics and proteomics data, samples from 2018 and 2020 were merged and normalised by a variation of RUV-III algorithm[44,118], implemented in the ruv 0.9.7.1 software package (https://CRAN.R-project.org/package=ruv), to adjust for unwanted variation and batch effects. RUV-III uses negative control proteins or metabolic, together with technical replicates, to estimate unwanted variation in the data arising from non-biological causes and removes it from data. The RUV-III approach exploits the fact that a number of patient samples had repeat profiles in both 2018 and 2020 (Supplementary Fig. 1). In the proteomics analysis, proteins that were missing in more than 75% of the samples in either 2018 or 2020 were filtered out. Proteins that had a sample variances less than the median and did not have any missing values in either 2018 or 2020 were used as negative control proteins in the normalisation. Samples from the same ventricle of the same patient were specified as technical replicates for RUV-III. The standard version of RUV-III does not allow missing values in the data, so a slight generalisation was implemented to allow the negative control effects to be estimated in an unbiased fashion. Our method is related to RUV-III-C (Poulos et al. 2020) but is not restricted to complete data only. In our analysis, a generalised averaging operator was estimated based on the abundances of negative control proteins in replicates and applied to each protein to obtain normalised abundances. For each protein, the generalised averaging operator is calculated by considering only the non-missing

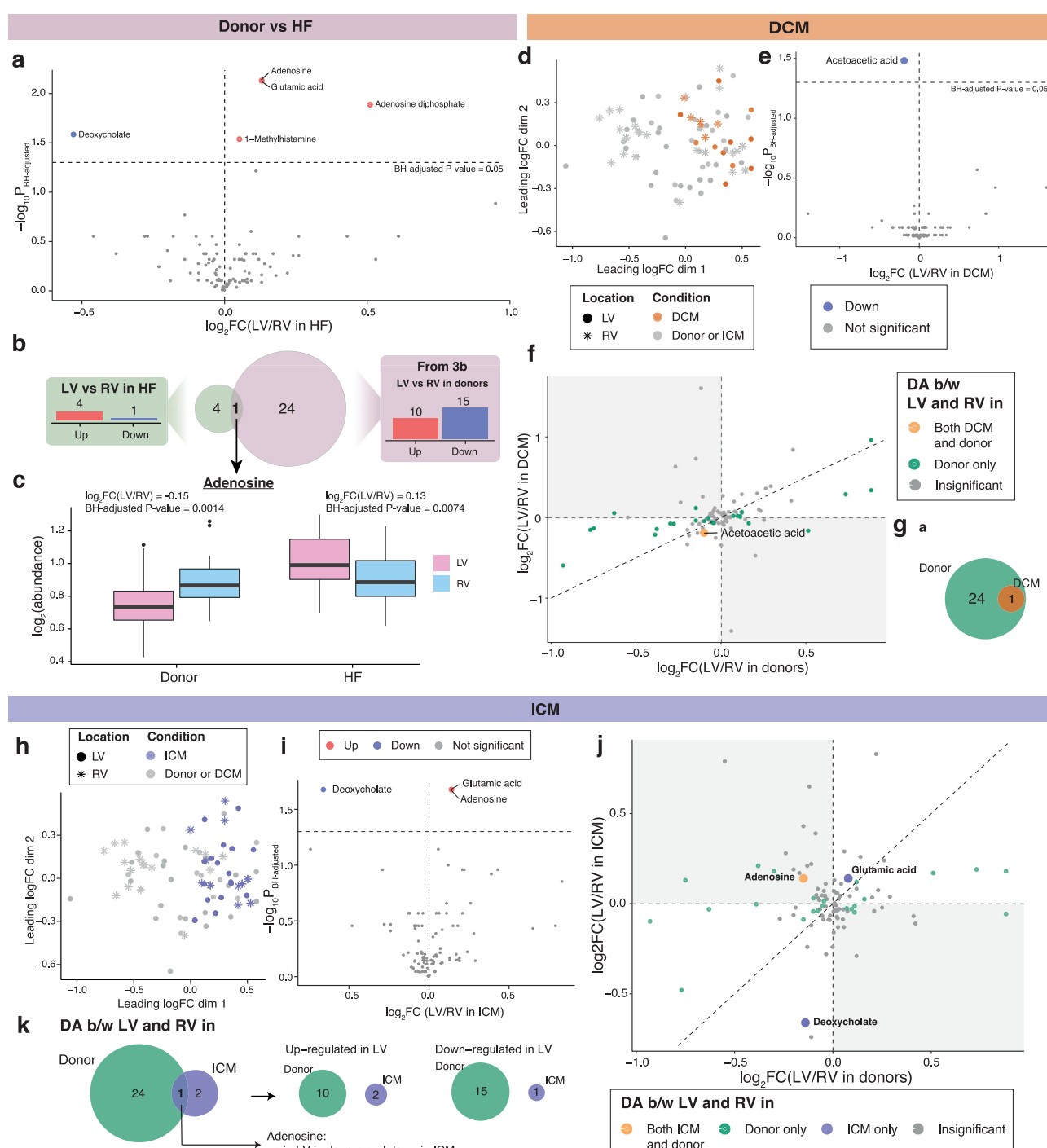

**Fig. 5 | Differential analysis in metabolomics abundance levels between ventricles in heart failures, DCM and ICM hearts compared with donors. a** Volcano plot of LV versus RV metabolites in heart failures. Estimates are derived using a linear regression model adjusted for sex and age, log2-transformed, with $n_{HF,LV} = 29$ and $n_{HF,RV} = 16$. Direction of regulation is in reference to RV, that is, up-regulated metabolites are more abundant in LV than in RV samples. **b** Venn diagram illustrating overlaps in DA metabolites between ventricles in donors and in HF, where the only DA metabolite common to donor and HF hearts is adenosine. **c** Boxplot of adenosine abundances in donors and HFs stratified by sample location. **d–g** Differential analysis in metabolite abundance levels between left and right ventricles in DCM hearts compared with donors, where $n_{DCM,LV} = 12$ and $n_{DCM,RV} = 5$, adjusted for sex and age, log2-transformed. Direction of regulation is in reference to RV. **d** MDS plot of metabolomics data with DCM samples highlighted in

brown. **e** Volcano plot of LV versus RV metabolites in DCM hearts. **f** Plot of log2 FCs of metabolites in LV versus RV samples in donors against those of DCM hearts. **g** Venn diagram counting the number of DA metabolites between ventricles shared by/unique to donors and/or DCM hearts. **h–k** Differential analysis in metabolite abundance levels between left and right ventricles in ICM hearts compared with donors, where $n_{ICM,LV} = 17$ and $n_{ICM,RV} = 11$, adjusted for sex and age, log2-transformed. Direction of regulation is in reference to RV. **h** MDS plot of metabolomics data with ICM samples highlighted in purple. **i** Volcano plot of LV versus RV metabolites in ICM hearts. **j** Plot of log2 FCs of metabolites in LV versus RV samples in donors against those of ICM hearts. **k** Venn diagrams counting numbers of all DA metabolites (left); up-regulated DA metabolites (middle); and down-regulated DA metabolites (right) between ventricles shared by/unique to donors and/or ICM hearts. Source data are provided in Supplementary Data 14, 20–25.

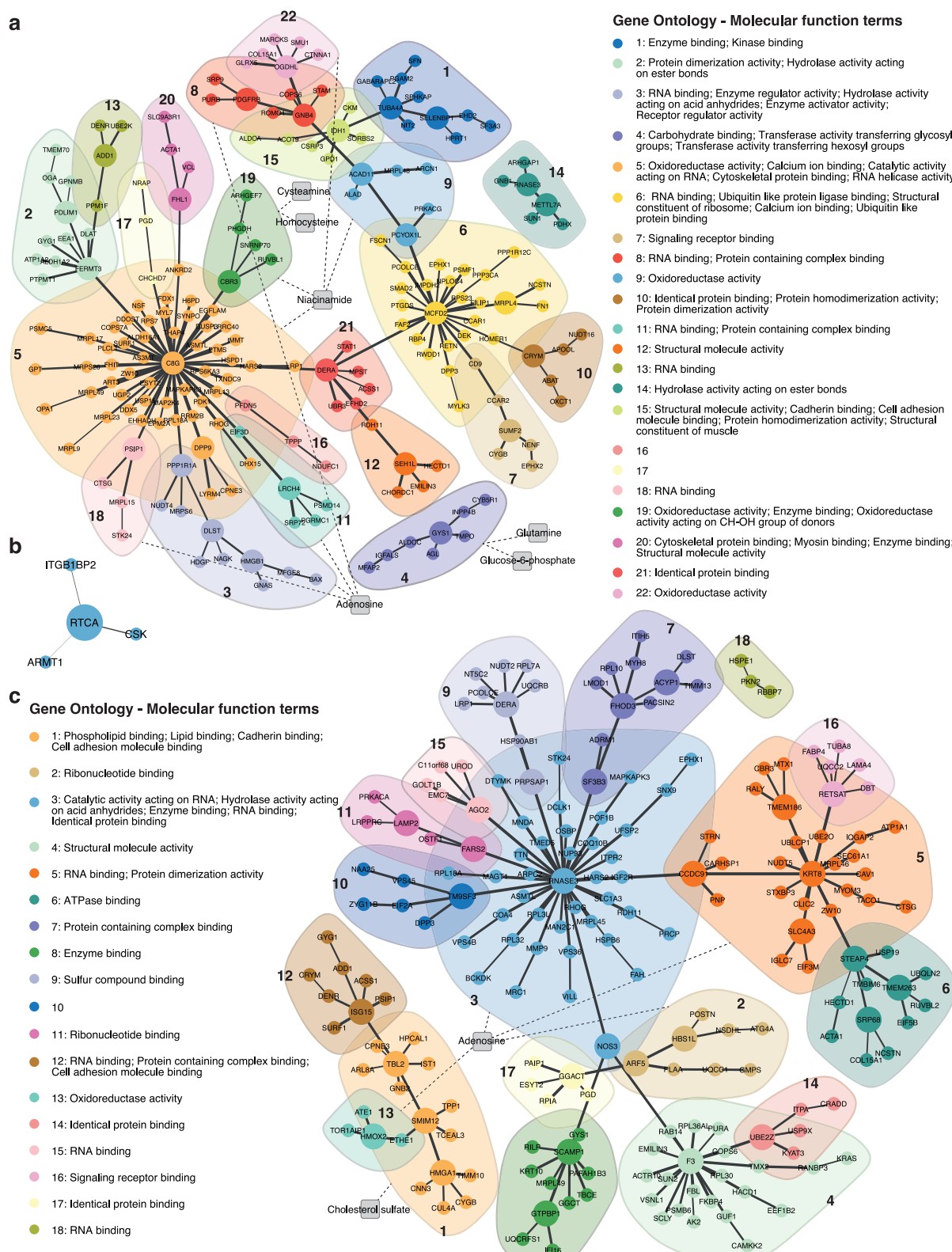

**Fig. 6 | DE protein (circles) correlation networks overlaid by potential biological links to DA metabolites (squares) in donor, DCM and ICM. a** Donor network plot. **b** DCM network plot. **c** ICM network plot. Proteomic data of top 200 DE proteins between ventricles are used to construct networks for donor and ICM. Protein nodes are coloured according to community detection results by the short random walk algorithm. Weights of edges between proteins (solid) are proportional to the absolute value of the pairwise Pearson correlation coefficient between proteins. Edges connecting metabolites to protein clusters (dashed) are annotated by hand, where for metabolite nodes, DA metabolites between ventricles for donor (BH-adjusted P-value ≤ 0.05), DCM or ICM (BH-adjusted P-value ≤ 0.1) are considered. Protein clusters are annotated by enriched Gene Ontology Molecular Function terms via gene set tests. Source data are provided in Supplementary Data 3, 9, and 12. (BH Benjamini-Hochberg; DA differentially abundant; DCM dilated cardiomyopathy; DE differential expression; ICM ischaemic cardiomyopathy).

**Article**

samples in the corresponding protein. More details can be found in Li (2020)[118]. In the metabolomics analysis, only metabolites measured in both 2018 and 2020 data were included. There were no missing values in the metabolomics data, so RUV-III with the default parameters was used. Sample variances were calculated for each metabolite and those with a sample variance less than the median were used as negative control metabolites in the normalisation. Samples from the same ventricle of the same patient were specified as technical replicates. After merging, only 13.4% of the proteomics data was missing and no metabolomics values were missing. Differential expression (DE) analysis for proteins and differential abundance (DA) analysis for metabolites were performed using the lmFit and eBayes functions in limma (3.48.3) R package[119]. Non-detected expression values were left as missing values for the expression analysis and were handled automatically by Limma. A linear model was fitted for each protein or metabolite to compare the protein expression levels or the metabolite abundance levels between ventricles in each condition (donors, DCM or ICM) while adjusting for sex and age for donors, DCM and ICM. A subject block factor was specified to account for individual heart effects via the duplicateCorrelation[120] function in limma. Pathways were annotated by KEGG[121]. KEGG pathway analysis was performed following DE analysis in proteins using kegga[122] from limma. Mean-rank gene set tests were performed based on the DA analysis of metabolites, where t-statistics from corresponding DA results were ranked using the geneSetTest[123] function in limma. In heatmaps, hierarchical clustering was performed using Euclidean distance as implemented in the pheatmap function from pheatmap R package (v1.0.12). Sex comparisons in donors in proteins and in metabolites were performed using the lmFit and eBayes functions in limma, where a linear model was fitted for each protein or metabolite after adjusting for age, and contrasts were specified to compare male with female samples at the same ventricle. The Benjamini-Hochberg (BH) method was used to calculate adjusted P-values to account for multiple comparisons in both DE and DA analyses.

MDS plots were generated by limma's plotMDS function. Protein networks were generated using the igraph (1.2.7) R package[124]. Protein nodes (circles) in each network are the top N DE proteins ranked by BH-adjusted P-value between ventricles in each condition, where $N = 200$ for donor and ICM; $N = 4$ for DCM. Weights of edges between proteins (solid) are proportional to the absolute value of the pairwise Pearson correlation coefficient between proteins. The graph layout was generated by Fruchterman-Reingold algorithm by the layout_with_fr function with default parameters in igraph together with the Minimal Spanning Tree (MST) and k-Nearest Neighbor (kNN) algorithm implemented by the mst.knn function with default parameters in mstknnclust (0.3.1) R package. The distance between every two proteins was determined by the inverse of the absolute value of their pairwise Pearson correlation coefficient. Protein clusters were identified via short random walks by the cluster_walktrap function with default parameters in igraph (where no cluster was detected for DCM). Gene Ontology (GO) Molecular Function (MF) terms (v7.4) were downloaded from MSigDB[125]. Protein clusters were annotated by enriched ($P$-value ≤ 0.05) GO MF terms via competitive gene set tests using limma[123]. Edges connecting proteins and metabolites (dashed) are curated by hand, where we manually examined each protein cluster with the DA metabolites (squares) between ventricles in donor or ICM.

## Data availability

Raw data and experimental conditions are available via online public repositories. The mass spectrometry proteomics data have been deposited to the ProteomeXchange Consortium (http://proteomecentral.proteomexchange.org) via the PRIDE[126] partner repository with the dataset identifiers PXD014826 for the 2018 dataset and PXD042155 for the 2020 dataset. The 2018 mass spectrometry proteomic raw data files can be linked to this study via Supplementary Data 30. The mass spectrometry metabolomics data have been deposited to the Metabolomics Workbench[127] under the project ID PR001684 (https://doi.org/10.21228/M81H71) with study IDs ST002716 (2018 dataset) and ST002717 (2020 dataset).

## Code availability

Code to reproduce the analyses presented here is available at https://github.com/Mengbo-Li/ruvms. as well as at Zenodo; DOI: 10.5281/zenodo.14031152[128].

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

## Acknowledgements

We thank the patients and staff of St Vincent's Hospital, Sydney, and the Australian Red Cross Blood Service. We thank Emeritus Prof. Cris dos Remedios and the late Dr. Victor Chang AC. We thank Ben Crossett and Stuart Cordwell from Sydney Mass Spectrometry. We thank Prof. Terence P. Speed for discussions on methods for data normalisation. This work was supported by the National Health and Medical Research Council (NHMRC) of Australia and the National Heart Foundation (NHF) of Australia. The contents of the published material are solely the responsibility of the individual authors and do not reflect the view of NHMRC or the NHF. This work was also supported by philanthropic donations to the University of Sydney and by a grant from the R.T. Hall Trust. Metabolomics Workbench raw data availability is supported by NIH grants U2C-DK119886 and OT2-OD030544 grants.

## Author contributions

B.H. performed sample preparation and extraction, data analysis and interpretation, produced the cellular schematic, and wrote the manuscript. M.L. conceived and performed statistical analyses under the supervision of G.K.S., designed and generated the figures, and helped write the manuscript. B.L.P. performed proteomic analyses. Y.C.K. performed metabolomic analyses and helped write the manuscript. D.H. performed proteomic analyses and helped write the manuscript. E.P. performed sample preparation and extraction. J.C. performed sample preparation and extraction. G.T.C. helped write the manuscript. O.G. performed sample preparation. G.K.S. supervised M.L. M.L. performed proteomic analysis. J.F.O.S. co-conceived the study, supervised the metabolomics, interpreted the data, and helped write the manuscript. S.L. co-conceived the study, oversaw the analyses, interpreted the data, and helped write the manuscript.

## Competing interests

The authors declare no competing interests.

## Ethical approval

This study was conducted in collaboration with multiple institutions across Australia, with contributions from researchers affiliated with the University of Sydney, the University of Melbourne, and St Vincent's Hospital, Sydney. All contributors fulfilling authorship criteria have been listed as co-authors, while those who provided additional support are acknowledged in the Acknowledgements section. The research was conducted with oversight from the Human Research Ethics Committee at the University of Sydney (USYD 2021/122). Sample procurement, including donor and heart failure tissue, and data collection and handling were performed in accordance with ethical guidelines. This research was designed to address globally relevant issues in heart failure and cardiomyopathy, and we have designed our sample cohort with this in mind where possible including heart disease prevalence. Additionally, we adhered to international best practices in data analysis and transparent reporting of our findings. All mass spectrometry data have been made available through public repositories, and we have ensured that local regulations concerning human tissue research were strictly followed.

## Informed consent

All tissue was collected following informed consent.

## Additional information

[1]Precision Cardiovascular Laboratory, The University of Sydney, Sydney, NSW, Australia. [2]Charles Perkins Centre, The University of Sydney, Sydney, NSW, Australia. [3]School of Medical Sciences, Faculty of Medicine and Health, The University of Sydney, Sydney, NSW, Australia. [4]Bioinformatics Division, The Walter and Eliza Hall Institute of Medical Research, Parkville, Victoria, VIC, Australia. [5]Department of Medical Biology, The University of Melbourne, Parkville, VIC, Australia. [6]Department of Anatomy and Physiology, The University of Melbourne, Parkville, VIC, Australia. [7]Heart Research Institute, Newtown, NSW, Australia. [8]School of Life and Environmental Sciences, Faculty of Science, The University of Sydney, Sydney, NSW, Australia. [9]Paediatric Oncology and Haematology, Oxford Children's Hospital, Oxford University Hospitals NHS Foundation Trust, Oxford, England. [10]Central Clinical School, Sydney Medical School, Faculty of Medicine and Health, The University of Sydney, Sydney, NSW, Australia. [11]Department of Cardiology, Royal Prince Alfred Hospital, Camperdown, NSW, Australia. [12]Kolling Institute, Royal North Shore Hospital, and Charles Perkins Centre, Faculty of Medicine and Health, University of Sydney, Sydney, NSW, Australia. [13]School of Mathematics and Statistics, The University of Melbourne, Parkville, VIC, Australia. [14]Faculty of Medicine, TU Dresden, Dresden, Germany. [15]The Baird Institute for Applied Heart and Lung Surgical Research, Sydney, NSW, Australia. [16]These authors contributed equally: Benjamin Hunter, Mengbo Li. [17]These authors jointly supervised this work: John F. O'Sullivan, Sean Lal. ✉e-mail: john.osullivan@sydney.edu.au; sean.lal@sydney.edu.au

