## [Transparent Peer Review file · Communications Biology]

Proteomic and metabolomic analyses of the human adult myocardium reveal ventricle-specific regulation in end-stage cardiomyopathies

Corresponding Author: Professor Sean Lal

This manuscript has been previously reviewed at another journal. This document only contains information relating to versions considered at Communications Biology.

Version 0:

Reviewer comments:

Reviewer #1

(Remarks to the Author)

Summary of methodology

- The proteomic sample preparation and quantification is largely aligned with MIAPE standards.
- The metabolomics section is well detailed as well.
- The stats are consistent with the field standard. LIMMA has been shown to be robust for proteomics.
- Description of the network tools used is good.
- Data has been deposited in databases.
- Tables have been more fully annotated.

As before, serving as reviewer 3, who did not review the original manuscript, the editors have asked me to evaluate the authors' responses to reviewer 2 and the revised manuscript. As such, my role is limited. I did have a few points of my own, last time, which the authors have addressed.

With respect to reviewer 2 point 2.

The crux of the disagreement between reviewer 2 and the authors appears to lie in how to best handle data collected on different instruments at different times. Reviewer 2 expresses concern that insights regarding LV vs RV could be impacted by use of data collected on different dates. This is a valid concern. The authors make valid points in rebuttal, however.

First, I think both reviewer 2 and the authors can agree that there is a large date/acquisition-dependent batch effect. The authors disclose this supplemental Fig 2A and 2B. So, from what I can see, the authors have been fully transparent.

I'm not a certified statistician, but I have extensive experience with omic analysis. I think we would all agree that there are several ways to mitigate the batch effect and extract biological insights. The authors describe two ways they have tried to guard against confounding date/instrument bias.

- The authors have done intra-date analyses from the two dates and compared analyses.
- They merged the two datasets, but specified batch ID as a covariate in the design matrix (Supplementary Figure G).

That second point is BINGO for me. It looks like the authors used the `ruv` R package. I probably would have set the study date as a factor in LIMMA and used the `removeBatchEffect` function. The two packages may work slightly differently, but either one should accomplish the task of minimizing the impact of the batch effect, irrespective of the historical source of the variance in the data. Specifically, any differences in the instrument, time to collection, homogenizer used, or room temperature, between the dates, ultimately manifests in the aggregate signal intensities, which culminates in a single date-effect on the data. Batch effect removal deals with this problem.

I suspect that reviewer 2 is essentially looking for a figure that explicitly illustrates before and after batch effect removal by

PCA or MDS for both the proteomics and the metabolomics, which the authors have provided in Supplementary Fig 2A and 2B.

The only other thing I can think of which could potentially isolate any date-dependent DE from chamber-dependent DE would be to take the batch-corrected data and then perhaps do a 1-way design, using 2018-LV, 2022-LV, 2018-RV and 2020-RV as groups, get the global p-value, and then plot a clustered heatmap. Any residual date-unique LV or RV clusters should reveal themselves and could be culled from the list.

But, again, if the batch removal does its job, which it looks like it has, the number of date-unique DE proteins/metabolites should be fairly small.

Re patient data disclosure.

To maximize the value of any resource the authors should endeavor to disclose as much as they are allowed under their institutional rules. My sense is the authors have done that.

Regarding the need for paired analyses.

Nice if you have it, as this is another way to diffuse the batch issue. But assuming the batch effect is handled in accordance with field standards, one could even argue that significant findings, on aggregate, are even more compelling than paired findings, as they are more robust (i.e. higher N values and significant in spite of interpersonal variation)

Reviewer 2 point 10 part 2

The normalization method shouldn't really be an issue. There are many widely used methods to normalize within datasets and just about all of them work reasonably well. I'm sure there's a paper out there that compared many of them and some may lend themselves to a marginal increase in statistical power. Nevertheless, the authors have chosen LOESS. Others might have chosen quantile normalization, but the authors' choice is reasonable. The point is that the same normalization method should be applied to each set. Then the linear model can validly determine the impact of date on the merged set, as described above for point #2.

Ultimately, the exact set of proteins that emerge as significant is nearly always influenced, to a degree, by the normalization method. Data points at the margin will hop from significant to non-significant. since every normalization affects the variance slightly differently. That's the reason studies must disclose the normalization method. The authors have done that.

Reviewer 3 (Me).

Point 1. Regarding centrifuging and losing most of the sarcomeres, cortical skeleton etc

We are unaware of evidence that suggests that our protein extraction methods result in a loss of sarcomeric and structural proteins.

The myofibrillar proteins are poorly soluble in 3% Triton X-100 (from personal experience), so I would have thought there was a decent chance they are still fairly insoluble in 4% deoxycholate. That said, only the authors know how big the 18,000g pellet is. If it's bigger than a speck, it contains the a substantial portion of the myofibrils. Of course, you would still identify many of the myofibrillar proteins, since they are also the most abundant in heart. So even if 95% pelleted, you would still identify them. The issue is that you may have only solubilized a subset of the myofibrils. In any case, it was a minor issue. Not worth belaboring.

Otherwise, the authors have addressed my issues.

REVIEWER COMMENTS

Reviewer #1 (Remarks to the Author):

The authors have satisfactorily addressed my previous queries and comments.

We thank the reviewer for their comments.

Reviewer #2 (Remarks to the Author):

First of all I want to state that I find the study subject highly interesting and highly relevant.

We thank the reviewer for their recognition of our work.

But I do not find that the authors have addressed the points raised sufficiently.

If this paper is considered as a resource, the authors should be

- 1) transparent on details of the samples analyzed
- 2) transparent on the how data was acquired for each of the samples

This has been improved with the information in supplementary tables 28 and 30. But this information is at the heart of the paper and should be more prominently shown.

We thank the reviewer for their comment, but we have provided all requested information in our first rebuttal, and as the reviewer has not specified what information is lacking, we are not able to know what they mean by transparency. Specifically:

- 1) We have provided all possible information regarding the samples/patients without breaching ethical (HREC) requirements.
- 2) In this rebuttal we have further expanded our methodology in response to the specific concerns of reviewer #2 and reviewer #3 (discussed where they have specified). Our methodology of data acquisition is not only provided in the methods of this manuscript but also in the database repositories, PRIDE and Metabolomics workbench.

We are grateful that the reviewer sees our manuscript as a highly relevant resource where we have integrated proteomic and metabolomic results into coherent biological processes and pathways.

From these new tables, one reads that measurements have been obtained from LV and RV samples collected from the same heart from 10 individuals in the present study, and from 4 (with 2 overlapping hearts) in the previously published data. These data represent the best data to evaluate for differences between LV and RV in non-diseased hearts, as all other samples are collected from different individuals and hence has inter-person differences in addition to LV and RV differences. Reviewer 1 eluded to the same in asking for a paired analysis. In the revision the authors did not convincingly address this point. Regardless of how good the analytical approach may be, there will be challenges due to 1) different mass spec measurement strategies and 2) inter-person differences when LV and RV are not collected from the same individual. The biological differences they can outline from the 10 hearts with paired LV and RV samples must be at the center of the analysis. And then they may show that they can improve the analysis by the strategies they apply. But a stringent analysis of the ten hearts with paired samples would be expected to be a subset of the findings they then make. From the response the provided to reviewer 1 that did not seem to be the case?

Reviewer 1 has stated that we have satisfied their questions/concerns.

We appreciate reviewer 2's concern of merging the datasets and have addressed this by performing new individual analyses for each dataset, which has been integrated into the manuscript as Supplementary Figures 4 and 5.

We are inclined to respectfully correct the reviewer that there are more than just '10' paired samples. We agree that information on paired samples could have been communicated more explicitly and thank the reviewer for bringing this to our attention. To remedy this, we have added the following in the Methods:

'This study featured 12 patients from the 2018 analyses (4 healthy donor, 4 DCM, and 4 ICM) and 21-22 patients from the 2020 analyses (11-12 healthy donor and 10 ICM) which contributed both LV and RV samples, resulting in 28-27 individuals (13-12 donor, 4 DCM, 11 ICM) individuals contributing both an LV and an RV sample across both cohorts of proteomic and metabolomic mass spectrometry analyses, respectively (supplementary Tables 28 and 30).'

As highlighted above, there are 28-27 individuals contributing paired samples in proteomic and metabolomic analyses, respectively.

I do not yet see analyses, where the authors convincingly show us that the differences they outline are truly reflecting biological differences between LV and RV, and rule out that they may be influenced by the different mass spec acquisition methods used in 2018 and 2020, the data normalization strategy applied or the biological differences between LV and RV donors. For this manuscript to be a resource, in my opinion, this is an important aspect to show.

These concerns have been addressed in detail in the points below.

To reaffirm, our thorough investigation has yielded biologically sound, complimentary, and plausibly effected metabolic pathways, as described in the Discussion, such that it would be highly unlikely that our results are due to, or strongly influenced by, differences in mass spectrometry acquisition. To illustrate our Discussion and to increase the impact of our manuscript, we have produced a cellular

schematic highlighting the (non-pathological) effected metabolic pathways in the donor LV compared to the RV in Supplementary Figure 1.

Below are responses to the original points raised.

Point 1:

Having the information provided in Supplementary Tables 28 and 30 are useful, and it remains my opinion that it is important to be as transparent as possible about the nature of the samples as it is critical for data evaluation. As I read Supplementary table 28, measurements from 69 samples were from the 2018 study and the new data included in this study was measurements from 50 samples, with some overlap as explained by authors. The authors write that they have incorporated novel and robust statistical methods in this study to merge datasets. I still have not seen sufficient analyses to really document this. The authors refer to Sup Fig. 1, but that by itself is not sufficient .

As a suggestion the authors could expand on the point of reviewer 1 following up on the analysis of the paired samples.

We thank the reviewer for their comment. As replied to (and kindly accepted by) Reviewer 1 in our previous response, the limitation in a paired analysis is that we need to account for the intra-subject correlation to make full use of all available information and avoid false discoveries, especially when we do not have access to paired samples in all subjects. In our results, we have used duplicate Correlation¹ function from the limma² R package. By accounting for the intra-subject correlation, we can estimate the mean-variance relationship in the proteins/metabolites with better precision, which improves the power of differential analysis whilst maintaining the false discovery control¹. Hereby instead, we have performed the within-batch analyses as mentioned in the following points as another means of evaluating the effectiveness of data merging. We thank the reviewer for the suggestion, and we have included these results as Supplementary Figures 4 (proteomics) and 5 (metabolomics), which we will discuss in more detail in the response below.

Point 2:

As a way to illustrate the power gained from combining the datasets, it would be reassuring to see an evaluation of the DE proteins identified in each of the 2018 and 2020 datasets analyzed separately (without the normalization strategy applied). The proteins identified in the separate analyses would be expected to be subsets of the DE set of proteins the authors find by their current strategy. Such a data representation would be reassuring in terms of ensuring that the normalization strategy is not driving interpretations.

We thank the reviewer for this suggestion. To address this, we have now performed the analyses using two alternative strategies that are often used for such an analysis, where we compare the proteomics (Supplementary Figure 4) and metabolomics (Supplementary Figure 5) expression in LV versus RV in healthy donor hearts by:

- (A) Within batch analysis as suggested by the reviewer, where each batch of data is analysed separately. This is to treat each batch as is and perform the analysis as one would if it had been a single-batch study. We do wish to point out that this within-batch strategy does not mean that we do not apply any normalisation on data, as one should still normalise within batch to remove other possible sources of unwanted variation, e.g., run order effects or random errors. Here we have applied cyclic loess as implemented by the `normalizeCyclicLoess` function in `limma`². As shown by Supplementary Figure 4B, this is an essential step. We then performed the differential expression analysis to compare LV vs RV samples in donor hearts within each batch while adjusting for log₂ age and sex as we did in the main results. The Venn diagram (4E) visualises the overlap in results obtained from merged data (as presented in Figure 2); 2018 data only and 2020 data only. We can see the majority of within-batch DE overlapped with those of merged data. The same can be shown in metabolomics data in Supplementary Figure 5. However, we also wish to point out that it is unrealistic to expect results from each within-batch analysis to be a complete subset of the merged dataset, especially in human samples; not just because the number of samples in each group (here, number of LV vs RV samples) are different, but also because we do not have exactly the same donors in both cohorts as we continuously recruit more samples. That most of the DE proteins identified within a batch also overlapped with results from the merged dataset further gives us confidence in the merged data.
- (B) Concatenating the two batches and include the batch ID as a covariate in the design matrix (Supplementary Figure G). In addition, we also included the individual heart ID in design to adjust for patient effects. We could not perform the `duplicateCorrelation` analysis as we did in the main results here due to the existence of across-batch replications. Nonetheless, including subject ID as a covariate is also an often used (but somewhat crude) strategy to adjust for individual effects. We compared results from concatenating the two batches with those of the merged dataset (Supplementary Figure 4G); and with those of within-batch analysis (Supplementary Figure 4H). We see a lot of overlap between the two strategies of combining the two cohorts. Also, importantly, there is a significant increase in statistical power because of the increased sample size by considering the two datasets simultaneously. The increased statistical power is also the result of a more balanced experimental design after combining the data, whereas we can see, the LV-RV ratio is not as balanced in 2018 as it is in 2020 data. Furthermore, we again see that most of the DE proteins by the within-batch approach were recovered by the concatenated analysis (Supplementary Figure 4H), with a few DE proteins that were uniquely identified by the within-batch analysis. Similar is true on metabolomics data (Supplementary Figure 5). Again, this further gives confidence in our original results.

Point 2i:

One has to be careful when merging data collected using different experimental acquisition strategies. I am sure the authors with the expertise they represent are well aware of this. For future readers of this work it is useful if the authors are more transparent on this subject.

We thank the reviewer for this comment. Details on the experimental acquisition strategy of both datasets are provided in the Methods section. While the power that can be gained by integrating data from different sources is huge, we agree that data merging/integration is a well-known challenge in large cohort and biobank studies for not only proteomics and metabolomics data, but RNA-seq and single-cell, etc. as well, especially when dealing with precious human samples that will not arrive in a strategically planned manner. We know the two batches were collected using different experimental acquisition strategies. However, they were both acquired by DIA methods; the number of overlapped proteins between the two batches after filtering by Q values and high missing percentage proteins is sufficiently high; and visualization QC diagnostics were reassuring, which has given us more confidence in merging these two datasets. We hereby thank the reviewer for the suggestion of more extensive QC and validation by alternative analysis strategies. As suggested by the reviewer, we have now included additional evaluation by analysing the two cohorts by alternative strategies as further QC and documented these results in supplementary figures for possible future reference by other groups who may wish to follow our methodology.

Point 2ii:

Are the authors not allowed to share any other data than age, sex and disease?

We thank the reviewer for their question. As mentioned in our first response, we are unable/not allowed to provide individual-level patient clinical data outside of age, sex, and disease phenotype, even if presented as being de-identified. Data must be provided in an aggregate fashion which is consistent with ours and other institutional ethics. We are limited in this regard due to our ethical obligations by receiving, storing, and publishing work with our donated human tissue. However, we can supply clinical data in aggregate form as already provided in Supplementary Table 1. This form of presentation was acceptable in our previous Nature Communications publication in 2020³.

Point 2iii):

Great that you share the raw data, as it is important for making full use of this as a resource.

We thank the reviewer for having brought this to our attention previously.

Point 3:

It is understandable if going into the mathematics of the strategy is beyond the scope of this paper. But it is the responsibility of the authors to convincingly show that their approach improves the outcome. This leads back to points 1 and 2.

We thank the reviewer for the comment.

Point 4:

I believe this would depend on the ethical permissions given. So back to point 2ii. The more information can be included the more useful the resource. But if these are the only information that can be shared that is understandable -yet the lack of details is a limitation.

We thank the reviewer for their comment and appreciate their request for more details. As mentioned, we are unable/not allowed ethically to present individual-level clinical data even if presented as apparently de-identified. However, we can present clinical data in aggregate form which had been supplied in Supplementary Table 1. We regret to remind the reviewer that we, unfortunately, do not have complete or equivalent clinical data for all patients, which is not uncommon for such human heart samples.

Point 5:

This is not supposed to be for me as a reviewer, but as a further strengthening of the study for future readers. So if you have histological data from all of the samples included, it would be valuable to include. Otherwise maybe you want to rephrase.

We thank the reviewer for their suggestion. We apologise for any miscommunication on our part. Our laboratory also operates the Sydney Heart Bank which cryogenically stores and preserves over 40,000 myocardial, aortic, vascular and plasma samples. The Sydney Heart Bank has been operating for ~35 years and it is the largest human heart bank in the world.

Histological verification was performed on fixed myocardium taken at the time of acquisition by the hospital where the surgery was taken place to confirm any disease pathology (formal anatomical histopathological analysis). Acquired tissue for the Sydney Heart Bank was immediately snap-frozen in liquid nitrogen within 40 minutes after aortic cross-clamp such that the myocardial tissue was not post-mortem.

In answering this question, we had realised that in our manuscript we had implied that the formal histological analyses confirmed whether the tissue was pre-mortem or not, as opposed to confirming any disease pathology. We have rectified this in the methods in the latest version of our manuscript. We again thank the reviewer for their comment and have also rephrased the manuscript text in reference to the above to be clearer to the reader.

We would like to take this opportunity to assure the reviewer of the quality of our donated myocardial tissue by referring them to additional previous publications which analysed our tissue, including a transcriptomics publication by Cell in 2019⁴⁻⁶:

Point 6:

Great that you share the data.

We thank the reviewer for having brought this to our attention previously.

Point 7:

It is great that you share the code used for the data analysis. Yet, even if the authors of this paper are experts in the field. It is still good scientific practice to provide information/analyses that document the robustness of the QC. Sup Fig.1 is not sufficient to do so.

We thank the reviewer for the comment. To this end, we have performed alternative analysis strategies as detailed in our responses to Points 1 and 2 as additional QC strategies. The results are now presented in Supplementary Figures 4 and 5. Code used for the additional analyses were also deposited into the GitHub repository.

Point 8:

Great that you share the code.

We thank the reviewer for having brought this to our attention previously.

Point 9:

The explanation provided is very good. My comment was meant as an invitation for the authors to elaborate in the manuscript text on what is presented in the various plots, such that future readers my better follow the rationale.

We thank the reviewer for having brought this to our attention previously and appreciate their suggestion. We have amended the Figure legends to add a short text to briefly describe the MDS plot where an MDS plot was presented.

Point 10:

This point has not really been addressed. The authors write that there is an even distribution of samples between datasets. But the 2020 dataset had 13 LV and 13 RV samples from donor hearts. The 2018 study had 24 LV samples and just 4 RV samples according to their Supplementary table 28. This is not an even distribution of samples.

We thank the reviewer for the correction and apologise for the unintended miscommunication. Yes, respective to paired samples; where an individual has contributed both an LV and RV sample in a dataset, the 2018 dataset featured less.

Regarding the origin of the reviewer's concern, the skewedness of the volcano plot of the donor proteomics data, we do not see the problem or what exactly needs to be addressed; the results between the ventricles produced a skewed appearance in the volcano plot. This is likely intrinsic to

the biological and developmental differences between the ventricles which does not weaken our manuscript. Furthermore, if the effects of disproportionate representation of proteins introduced by our methodology or unequal pairs between datasets was causative to the skewed plot this would have also been evidenced elsewhere, specifically the MDS plots. This result is not an indication of fault in our methodology or our analyses. As the reviewer's concerns appear to arise from scepticism of our data merger, we hope our new alternative analyses (Supplementary Figures 4 and 5) demonstrating successful dataset merger will satisfy this matter.

Point 10 part 2:

The authors write that the normalization strategy was successful, which is great. For concluding this, I would still like to see outcome of an analysis that shows that deeming proteins DE is not driven by the normalization strategy. If you analyze each of the datasets independently, without the normalization strategy, are the proteins found DE in those approaches a subset of what you get with this improved method.

We thank the reviewer for their comment. We agree with their reasoning and, therefore, performed new differential analyses on both the proteome and the metabolome for each dataset individually and compared the results to the merged analyses. The results of our new analyses and their comparison to our merged dataset can be found in Supplementary Figures 4 (protein) and 5 (metabolites).

As the reviewer may empathise with the potential implications of performing an alternative validation analysis; competing with and undermining the innovation and results of our merged dataset, potentially confusing readers as to why an alternative analysis was done, and in order not to communicate a lack of confidence in our own work, we deliberately did not supply an entirely new output of individual proteins/metabolites (matrices/Supplementary tables) alongside our current merged data.

Point 15:

I agree that the authors have generated a dataset that can hold great value. But it should be the authors responsibility to confirm the findings they claim. There are limitations in the study design - which the authors can certainly argue for as they are working with precious and limited human material -nonetheless, given such limitations necessitates that the findings are confirmed.

We thank the reviewer for their comment. As mentioned previously, we believe we have been conservative with our claims such that we have not inappropriately confirmed any finding outside the scope of our analyses. In our study, we had suggested many differences in the proteome and metabolome of the healthy human adult left ventricle compared to the right ventricle, as an example. The limitations of our study are clearly outlined, as observed by the reviewer. Our limitations are as expected in a spatially unbiased mass spectrometry analysis using human primary tissue and do not invalidate our findings. The language used in the Results and Discussion is in

reflection of this and in-line with other multi-omics publications in high-impact journals. Respectively, the reviewer has made no mention as to what findings should be confirmed nor what findings they take issue with. We do not believe we should rationally be expected to perform translational experiments on potentially hundreds of proteins or metabolites. Furthermore, we are unaware of quantitative techniques assessing many molecules with greater accuracy than mass spectrometry, which is obviously why we chose this technique for this study.

This study is large in scope, identifying many avenues for future research. However, we have also performed a considerably thorough investigation, unlike any undertaken before. Our analysis was not to interrogate a single targeted disease mechanism, but to characterise the adult human left ventricular myocardium compared to the right ventricle.

We would kindly like to correct the reviewer of any misinterpretation of our previous statements; we have not argued that working with limited human material should allow us an exemption to excellent scientific rigor. Rather, our manuscript and previous responses have attempted to communicate that this study far exceeds any before it; despite human tissue being notoriously scarce, we examined pre-mortem human myocardium from 60 individuals devoted solely on an inter-ventricular analysis. The importance of this manuscript and how it will greatly contribute to the fields of biological science and translational medicine was reflected in the previous response from the reviewer where they were unable to find a comparable study.

Point 16:

The authors argue that their study is focused on comparisons between LV and RV, which is great. But it seems somewhat misleading that they keep pointing out that they have analyzed 60 samples in total, when they have data from 10 donor hearts where both LV and RV samples were collected. I do not in any way mean to neglect the value nor the importance of the data collected. But the authors could do an effort in presenting what they have in a more transparent manner and be more reflective of the work done by others.

We thank the reviewer for their comment. In the previously supplied Supplementary Tables 28 and 30, readers can fully appreciate what samples were used in all analyses. We appreciate that the point the reviewer has raised is an important one and so we have amended the Methods in Acquisition of Human Myocardium:

'This study featured 12 patients from the 2018 analyses (4 healthy donor, 4 DCM, and 4 ICM) and 21-22 patients from the 2020 analyses (11-12 healthy donor and 10 ICM) which contributed both LV and RV samples, resulting in 28-27 individuals (13-12 donor, 4 DCM, 11 ICM) individuals contributing both an LV and an RV sample across both cohorts of proteomic and metabolomic mass spectrometry analyses, respectively (supplementary Tables 28 and 30).'

Given the information presented above which was previously provided in the form of Supplementary Tables 28 and 30, we would like to respectively correct the reviewer that not just 10 donor hearts contributed both LV and RV samples.

Reviewer #3 (Remarks to the Author):

As reviewer 3, who did not review the original manuscript, the editors have asked me to evaluate the authors' responses to reviewer 2 and the revised manuscript. As such, my role is limited and I will not be providing a detailed independent review with too many new suggestions, as this might be a tad unfair to the authors.

In any case, reviewer 2 was indeed thorough and the bulk of the critiques were sound. He/she raised a few issues.

1. There were genuine concerns that the merger of datasets from different dates on conducted different instruments is often problematic. Supplementary figure 1, would seem to do a reasonable job of addressing this. It is interesting though that the signal distribution of the 2020 proteome set remains a bit broader/skewed even after normalizing both datasets to a common mean. I'm somewhat surprised that this isn't reflected in the merged MDS (why MDS, not PCA?) Nevertheless, on the basis of the merged MDS, it would seem that systemic/date batch effects are eliminated.

We thank the reviewer for the comment. Conceptually, MDS is equivalent to PCA when using Euclidean distances (and we are), but MDS allows for missing values in data whereas PCA does not. In the MDS plot, distances on the plot represent the leading log₂-fold-changes. The leading log₂-fold-change between a pair of samples is defined as the root-mean-square average of the top largest log₂-fold-changes between those two samples. The PCA plot uses the same genes throughout (which is why no missing values are allowed) whereas the MDS plot potentially selects different genes to distinguish each pair of samples. It often gives better resolution than a PCA plot if different molecular pathways are relevant for distinguishing different pairs of samples, especially when missing values are present. Alternatively, one can impute the data to get a complete data matrix to generate the PCA plot. However, we did not take this approach here as firstly, there was only a modest number of missing values left in data after quality control; and secondly, careless imputation is prone to biased results, as we recently showed in Li and Smyth (2023)⁷.

As pointed out, in the normalised RLA plot, some 2020 samples are more variable than the rest. It is not necessarily worrying though; a perfectly aligned RLA plot would only appear when the data have absolutely no differential expression. Also as pointed out, we performed other diagnostic plots such as MDS/PCA in complement, which have assured us the merging has been effective.

2. Reviewer 2 had concerns about which samples were compared (how many reanalyses, were analyses paired etc.). The authors have addressed this with new supplementary tables.

We appreciate the reviewer's comment.

3. Reviewer 2 was nonplussed by the authors' initial failure to deposit the proteomic and metabolomic data into public repositories. I concur with reviewer2, that this really is an issue, as the reviewer can't actually independently evaluate the data otherwise. This really should be a journal requirement prior evaluation of the manuscript. In any case, the authors have rectified the issue.

We appreciate the reviewer's comment.

4. Reviewer 2 seemed to be requesting patient-by-patient medical workup. I understand the motivation as it could help identify sources of variance or co-variables. Nevertheless, I don't necessarily see this a major issue.

We appreciate the reviewer's understanding.

5. Reviewer 2 took issue with the authors' assessment that their study was the most comprehensive of its kind. With respect to proteome depth, reviewer 2 is right. This study doesn't come close to those previously published by Mann and Lundby groups. However, in their rebuttal, the authors lay out the distinctions of their study, which are not inconsequential. In fact, it is worth noting some of those distinctions in the discussion to help distinguish their study. This would also satisfy reviewer 2's desire that the study should be better contextualized relative to prior studies.

We thank the reviewer for their comment and suggestion. We have amended the discussion of our manuscript to better contextualise and separate our study from others.

Before I note a few of my own issues, this is a quite well-performed multi-omic study that makes a fairly solid contribution to the field in the same way that all good multi-omic papers do. It provides tantalizing leads that ultimately need to be tested experimentally. There's nothing wrong with that. Not necessarily a paradigm shift, in and of itself, though. Incidentally, I would have said the same thing about the aforementioned studies by the Mann and Lundby groups. The principal significance comes from the fact that the samples are human and premortem. The statistical analysis is sound.

Minor issues.

1. As described, the authors have actually analyzed a sarcomerically- and structural protein-depleted cardiac proteome (i.e. sonicating tissue powder in a buffer with detergent and centrifuging at 18,000xg removes the vast bulk of the myofilaments, cortical cytoskeleton, and fibrous extracellular matrix proteins). This doesn't invalidate the study, but it should be noted somewhere, either in the methods section or in the limitations section of the discussion.

We thank the reviewer for their comment and suggestion. We are unaware of evidence that suggests that our protein extraction methods result in a loss of sarcomeric and structural proteins. Indeed, we have quantified these proteins, so we believe our analyses are relevant. As all samples of a dataset were exposed to the same procedures at the same time, and that there were no notable differences between the two datasets after normalisation (no unequal representation), we feel it is difficult to suggest that sarcomeric and structural proteins were indeed depleted in the samples due to our methodology. Due to this, we have respectfully decided to leave out this indeterminate limitation.

2. The LC portion of the methods is a bit sparse. A prior paper is referenced but it would be optimal if the LC solvent system was stated more explicitly. I'm assuming A and B solvents had formic acid in them?

We thank the reviewer for their comment. We have added more details to the methods of the manuscript regarding A and B solvents.

3. On line 435, I was confused. What high pH fractions? I assumed this was standard acidic RP-HPLC, no?

We thank the reviewer for their comment. We have added more details to the methods of the manuscript regarding peptide fractionation.

4. The hierarchical clustering method isn't described. What distance metric was used? What was the linkage criterion?

We thank the reviewer for this comment. The following text was added to Methods section:

"In heatmaps, hierarchical clustering was performed using Euclidean distance as implemented in the pheatmap function from pheatmap R package (v1.0.12)."

5. The heatmap and a few figures use green and red, which are not color-blind-friendly.

We thank the reviewer for the comment. We presume it was referring to the pink and khaki colours for LV and RV coding. We have updated the colour legend for LV/RV to avoid using green and red together for the same variable.

6. Annotation of the supplementary tables is fairly sparse. Each table is indistinguishable from the rest. The authors have taken care to make appealing figures, they could afford to give the supplementary tables the same TLC.

We thank the reviewer for their comment and have amended and reformatted the Supplementary Tables to be easier to navigate.

References:

1. Smyth, G.K., Michaud, J.I. & Scott, H.S. Use of within-array replicate spots for assessing differential expression in microarray experiments. *BIOINFORMATICS* **21**, 2067-2075 (2005).
2. Ritchie, M.E., *et al.* limma powers differential expression analyses for RNA-sequencing and microarray studies. *Nucleic Acids Research* **43**(2015).

3. Li, M.B., *et al.* Core functional nodes and sex-specific pathways in human ischaemic and dilated cardiomyopathy. *Nat. Commun.* **11**(2020).
4. Mollova, M., *et al.* Cardiomyocyte proliferation contributes to heart growth in young humans. *Proceedings of the National Academy of Sciences of the United States of America* **110**, 1446-1451 (2013).
5. Polizzotti, B.D., *et al.* Neuregulin stimulation of cardiomyocyte regeneration in mice and human myocardium reveals a therapeutic window. *Sci. Transl. Med.* **7**, 13 (2015).
6. van Heesch, S., *et al.* The Translational Landscape of the Human Heart. *Cell* **178**, 242-+ (2019).
7. Li, M. & Smyth, G.K. Neither random nor censored: estimating intensity-dependent probabilities for missing values in label-free proteomics. *Bioinformatics (Oxford, England)* **39**(2023).

Response to the editor (in *Blue*)

We now invite you to revise your paper one last time to address the remaining concerns of our reviewer. Please consider their suggestion regarding the isolation of any date-dependent DE from chamber-dependent DE. In addition, please ensure that any relevant papers related to your study, including for instance any papers which your previous proteomic datasets originate from (eg PMID: 32487995), are adequately discussed in the introduction section.

Dear Dr. Dario Ummarino,

Thank you greatly for your invitation to review our manuscript and supporting documentation one last time. We have addressed all concerns; however, we have respectfully decided that it is not in the best interest of our manuscript to perform additional analyses in the form of a date-dependent DE analysis as we believe we have already satisfied the concern arising from this and that it would confuse the reader.

The limma removeBatchEffect approach suggested by Reviewer 3 is equivalent to an analysis already presented in Supplementary Figure 4 with year included a factor in the design matrix. One of us (Professor Smyth) is the author of the limma and removeBatchEffect tools, so is well-qualified to comment on the equivalence of the different approaches. The RUV-III normalization method used in the Results section is far more comprehensive and reliable in this case, however, because it incorporates all the available information from both years (including cross year replicates and negative control proteins) into the normalization and because it removes multiple sources of variation rather just the year effect. RUV does not require the source of unwanted variation to be pre-specified as does removeBatchEffect. It is well established that omics data collected on human subjects over time typically show a lot of background variation and RUV is frequently able to improve data consistently, see for example Poulos et al (Nature Communications 2020) and Molania et al (Nature Biotechnology 2023).

Poulos, R.C., Hains, P.G., Shah, R. et al. Strategies to enable large-scale proteomics for reproducible research. Nat Commun 11, 3793 (2020). <https://doi.org/10.1038/s41467-020-17641-3>

Molania, R., Foroutan, M., Gagnon-Bartsch, J.A. et al. Removing unwanted variation from large-scale RNA sequencing data with PRPS. Nat Biotechnol 41, 82–95 (2023). <https://doi.org/10.1038/s41587-022-01440-w>

Response to the Reviewer (in *Blue*)

REVIEWERS' COMMENTS:

Reviewer #3 (Remarks to the Author):

Summary of methodology

- The proteomic sample preparation and quantification is largely aligned with MIAPE standards.
- The metabolomics section is well detailed as well.
- The stats are consistent with the field standard. LIMMA has been shown to be robust for proteomics.
- Description of the network tools used is good.
- Data has been deposited in databases.
- Tables have been more fully annotated.

As before, serving as reviewer 3, who did not review the original manuscript, the editors have asked me to evaluate the authors' responses to reviewer 2 and the revised manuscript. As such, my role is limited. I did have a few points of my own, last time, which the authors have addressed.

With respect to reviewer 2 point 2.

The crux of the disagreement between reviewer 2 and the authors appears to lie in how to best handle data collected on different instruments at different times. Reviewer 2 expresses concern that insights regarding LV vs RV could be impacted by use of data collected on different dates. This is a valid concern. The authors make valid points in rebuttal, however.

First, I think both reviewer 2 and the authors can agree that there is a large date/acquisition-dependent batch effect. The authors disclose this supplemental Fig 2A and 2B. So, from what I can see, the authors have been fully transparent.

I'm not a certified statistician, but I have extensive experience with omic analysis. I think we would all agree that there are several ways to mitigate the batch effect and extract biological insights. The authors describe two ways they have tried to guard against confounding date/instrument bias.

- The authors have done intra-date analyses from the two dates and compared analyses.
- They merged the two datasets, but specified batch ID as a covariate in the design matrix (Supplementary Figure G).

That second point is BINGO for me. It looks like the authors used the `ruv` R package. I probably would have set the study date as a factor in LIMMA and used the "removeBatchEffect" function. The two packages may work slightly differently, but either one should accomplish the task of minimizing the impact of the batch effect, irrespective of the historical source of the variance in the data. Specifically, any differences in the instrument, time to collection, homogenizer used, or room temperature, between the dates, ultimately manifests in the aggregate signal intensities, which culminates in a single date-effect on the data. Batch effect removal deals with this problem.

I suspect that reviewer 2 is essentially looking for a figure that explicitly illustrates before and

after batch effect removal by PCA or MDS for both the proteomics and the metabolomics, which the authors have provided in Supplementary Fig 2A and 2B.

The only other thing I can think of which could potentially isolate any date-dependent DE from chamber-dependent DE would be to take the batch-corrected data and then perhaps do a 1-way design, using 2018-LV, 2022-LV, 2018-RV and 2020-RV as groups, get the global p-value, and then plot a clustered heatmap. Any residual date-unique LV or RV clusters should reveal themselves and could be culled from the list.

But, again, if the batch removal does its job, which it looks like it has, the number of date-unique DE proteins/metabolites should be fairly small.

We thank the Reviewer for their understanding and suggestion. We have respectfully decided that it is not in the best interest of our manuscript to perform additional analyses in the form of a date-dependent DE analysis as we believe we have already satisfied the concern arising from this and that it would confuse the reader.

The limma removeBatchEffect approach suggested by Reviewer 3 is equivalent to an analysis already presented in Supplementary Figure 4 with year included a factor in the design matrix. One of us (Professor Smyth) is the author of the limma and removeBatchEffect tools, so is well-qualified to comment on the equivalence of the different approaches. The RUV-III normalization method used in the Results section is far more comprehensive and reliable in this case, however, because it incorporates all the available information from both years (including cross year replicates and negative control proteins) into the normalization and because it removes multiple sources of variation rather just the year effect. RUV does not require the source of unwanted variation to be pre-specified as does removeBatchEffect. It is well established that omics data collected on human subjects over time typically show a lot of background variation and RUV is frequently able to improve data consistently, see for example Poulos et al (Nature Communications 2020) and Molania et al (Nature Biotechnology 2023).

Poulos, R.C., Hains, P.G., Shah, R. et al. Strategies to enable large-scale proteomics for reproducible research. Nat Commun 11, 3793 (2020). <https://doi.org/10.1038/s41467-020-17641-3>

Molania, R., Foroutan, M., Gagnon-Bartsch, J.A. et al. Removing unwanted variation from large-scale RNA sequencing data with PRPS. Nat Biotechnol 41, 82–95 (2023). <https://doi.org/10.1038/s41587-022-01440-w>

Re patient data disclosure.

To maximize the value of any resource the authors should endeavor to disclose as much as they are allowed under their institutional rules. My sense is the authors have done that.

We thank the Reviewer for their comment and understanding.

Regarding the need for paired analyses.

Nice if you have it, as this is another way to diffuse the batch issue. But assuming the batch effect is handled in accordance with field standards, one could even argue that significant

findings, on aggregate, are even more compelling than paired findings, as they are more robust (i.e. higher N values and significant in spite of interpersonal variation)

We thank the Reviewer for their comment and understanding.

Reviewer 2 point 10 part 2

The normalization method shouldn't really be an issue. There are many widely used methods to normalize within datasets and just about all of them work reasonably well. I'm sure there's a paper out there that compared many of them and some may lend themselves to a marginal increase in statistical power. Nevertheless, the authors have chosen LOESS. Others might have chosen quantile normalization, but the authors' choice is reasonable. The point is that the same normalization method should be applied to each set. Then the linear model can validly determine the impact of date on the merged set, as described above for point #2.

Ultimately, the exact set of proteins that emerge as significant is nearly always influenced, to a degree, by the normalization method. Data points at the margin will hop from significant to non-significant. since every normalization affects the variance slightly differently. That's the reason studies must disclose the normalization method. The authors have done that.

We thank the Reviewer for their comment and understanding.

Reviewer 3 (Me).

Point 1. Regarding centrifuging and losing most of the sarcomeres, cortical skeleton etc We are unaware of evidence that suggests that our protein extraction methods result in a loss of sarcomeric and structural proteins.

The myofibrillar proteins are poorly soluble in 3% Triton X-100 (from personal experience), so I would have thought there was a decent chance they are still fairly insoluble in 4% deoxycholate. That said, only the authors know how big the 18,000g pellet is. If it's bigger than a speck, it contains the a substantial portion of the myofibrils. Of course, you would still identify many of the myofibrillar proteins, since they are also the most abundant in heart. So even if 95% pelleted, you would still identify them. The issue is that you may have only solubilized a subset of the myofibrils. In any case, it was a minor issue. Not worth belaboring.

Otherwise, the authors have addressed my issues.

We thank the Reviewer for alerting us to this. This is something we should keep an eye on in the future as it was something we were not aware of before. We thank the Reviewer for their comment and for sharing their knowledge on this. The pellet was bigger than a spec and was such for all the sample tubes.